# Pose Splatter: A 3D Gaussian Splatting Model for Quantifying Animal Pose and Appearance

**Jack Goffinet**[*,1], **Youngjo Min**[*,1], **Carlo Tomasi**[1], **David E. Carlson**[1,2,3]

[1] Department of Computer Science
[2] Department of Biostatistics and Bioinformatics
[3] Department of Civil and Environmental Engineering
Duke University
Durham, NC 27705
{jack.goffinet, youngjo.min, tomasi, david.carlson} @duke.edu,

## Abstract

Accurate and scalable quantification of animal pose and appearance is crucial for studying behavior. Current 3D pose estimation techniques, such as keypoint- and mesh-based techniques, often face challenges including limited representational detail, labor-intensive annotation requirements, and expensive per-frame optimization. These limitations hinder the study of subtle movements and can make large-scale analyses impractical. We propose *Pose Splatter*, a novel framework leveraging shape carving and 3D Gaussian splatting to model the complete pose and appearance of laboratory animals without prior knowledge of animal geometry, per-frame optimization, or manual annotations. We also propose a rotation-invariant visual embedding technique for encoding pose and appearance, designed to be a plug-in replacement for 3D keypoint data in downstream behavioral analyses. Experiments on datasets of mice, rats, and zebra finches show *Pose Splatter* learns accurate 3D animal geometries. Notably, *Pose Splatter* represents subtle variations in pose, provides better low-dimensional pose embeddings over state-of-the-art as evaluated by humans, and generalizes to unseen data. By eliminating annotation and per-frame optimization bottlenecks, *Pose Splatter* enables analysis of large-scale, longitudinal behavior needed to map genotype, neural activity, and behavior at high resolutions.

## 1 Introduction

The study of animal behavior is a central focus in neuroscience research, as it provides essential context for understanding neural and physiological processes. In particular, accurate capture of 3D pose allows researchers to study important elements of animal behavior including walking, balance, and interaction with the environment [38]. These elements are essential for detecting small behavioral changes associated with neurological diseases or therapies [62].

Deep learning methods originally developed for 3D human pose estimation from images [4, 46, 21] have inspired and driven dramatic advances in 3D animal shape and pose reconstruction as well [83, 45, 25, 75, 1, 5]. These methods are most often based on either keypoints or meshes.

Keypoint-based methods [45, 25, 12, 18, 14] are straightforward to implement and relatively efficient computationally. They triangulate a set of anatomical points (e.g., joints, wing edges, or tail tips) across multiple camera views to reconstruct their 3D positions. However, each keypoint must be

---

[*]These authors contributed equally to this work.

39th Conference on Neural Information Processing Systems (NeurIPS 2025).

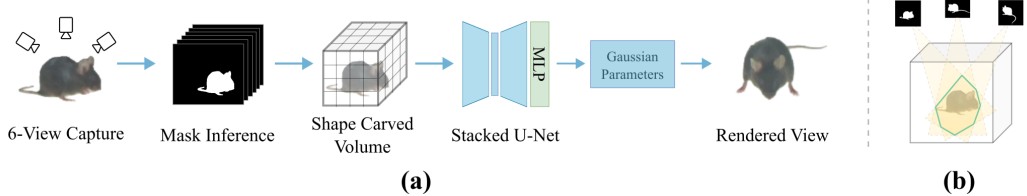

**(a)**                                            **(b)**

Figure 1: **(a)** *Pose Splatter* **pipeline.** Multi-view images and their corresponding masks are carved into a coarse voxel shape, which a stacked U-Net converts into de-voxelized 3D Gaussian parameters that are finally rendered through Gaussian splatting. The entire process runs in only 2.5 GB of GPU memory (VRAM) compared to 10-20 GB for competing methods. **(b) Shape-carving concept.** Silhouettes from each camera are back-projected into a shared voxel grid (yellow cones), removing voxels outside the visual hull. The green intersection marks the rough volumetric prior fed to the network.

located accurately for training, leading to labor-intensive annotation. Additionally, these landmarks are too sparse to capture the body's full geometry, let alone the color and texture of its surface.

Mesh-based methods [83, 82, 71, 66, 1, 5, 64, 80, 33, 54] do reconstruct the animal's complete surface geometry by fitting a parameterized 3D model to observed data, and thereby facilitate in-depth analyses of body shapes and deformations. However, inferring this richer information requires specialized and time-intensive mesh-fitting routines for each input frame. Inference often also depends on an accurate template model (e.g., SMAL [83]) and may fail to generalize to postures that are not well-represented by the original template, or to altogether different species (e.g., mice, rats).

To address these challenges, we propose *Pose Splatter*, a feed-forward model based on 3D Gaussian splatting (3DGS) that reconstructs 3D shapes of different animal species accurately and in a scalable way. 3DGS [26] is a 3D scene rendering technique that has recently become popular due to its extremely fast rendering times and high visual fidelity. It represents a scene as a collection of Gaussian particles, each with its own position, covariance, and color parameters. While earlier models optimized these parameters on a scene-by-scene basis, recent efforts have introduced feed-forward variants [10, 11] designed to perform single-step inference after training. However, unlike *Pose Splatter*, these methods assume ample inter-view overlap and therefore struggle with sparse-view 3D animal reconstruction, where such overlap is minimal.

*Pose Splatter* captures each scene with a small set of calibrated cameras (4–6 in this work) and generates foreground masks for each view using SAM2 [51]. A shape-carving process [31, 30] back-projects these multi-view masks into 3D visual cones, whose intersection yields a rough estimate of the voxels inside the animal's shape. A stacked 3D U-Net then refines this coarse volume, and a compact multi-layer perceptron (MLP) processes the voxel-level features to produce the parameters for the 3D Gaussian splats, which are rendered with 3DGS. The network is trained end-to-end to minimize image-based losses. We additionally introduce a visual embedding technique that relies on the ability to render the scene from novel viewpoints and provides an informative low-dimensional descriptor of pose and appearance for use in downstream analyses.

Our framework addresses several limitations of existing techniques. Unlike keypoint-based approaches, which use a sparse representation of body and require extensive landmark annotation, our framework recovers the **complete 3D posture of the animal without any manual labeling**. In contrast to mesh-based methods, which often demand per-frame optimization and accurate template models, our network **performs inference via a single forward pass and requires no species-specific templates**. Our quantitative and qualitative experiments demonstrate that our model achieves robust 3D reconstructions and novel view renderings of multiple animal species despite its simplicity. Our experiments show that the proposed **visual embedding captures subtle variations in animal pose and serves as a useful descriptor for behavioral analysis.** By removing manual annotation and computational bottlenecks, our approach opens the door to large-scale and high resolution behavioral analysis, facilitating deeper understanding of the genetic and neural underpinnings of behavior.

## 2 Related Work

**Keypoint-based Pose Estimation**  Keypoint-based pose estimation research first concentrated on 2D keypoint estimate for human pose [8, 49, 74, 16, 55, 56, 58, 77], where models aimed to localize joints in 2D images to infer human poses. The field moved toward 3D keypoint estimation [41, 52, 3, 22, 29, 6, 21, 50], using multi-view information to represent human poses in a more realistic, spatially aware manner. This evolution was mirrored in animal pose estimation, with early studies primarily tackling 2D keypoint estimation [47, 19, 7, 42, 32, 73, 40, 53] and more recent studies shifting to 3D keypoint estimation [45, 2, 76, 75, 37, 12], allowing for a more accurate representation of animal poses in three-dimensional space. However, these models capture only a sparse representation and depend on manually annotated training data. *Pose Splatter*, by contrast, reconstructs the animal's complete 3D geometry and requires no manual labels.

**Mesh-based Pose Estimation**  Mesh-based 3D pose estimation provides a more complete representation of body shape and surface characteristics compared to keypoint methods by fitting a parametric 3D model to observed data. Mesh models have been extensively applied in human pose estimation [34, 4, 20, 24, 28, 13] to capture fine-grained details, such as muscle contours and body surface deformations, thereby enabling them especially in applications requiring great accuracy, like biomechanics and animation. Mesh-based techniques [83, 81, 71, 69, 66, 1, 5, 59, 64, 54] have recently been used in animal pose estimation, providing a more detailed representation of the diverse anatomies and movements of different species compared to keypoint methods. Unlike mesh methods, *Pose Splatter* does not require a species-specific template or per-frame optimization routines.

**3D Gaussian Splatting (3DGS)**  3DGS [26] has emerged as a powerful technique for novel view synthesis and 3D reconstruction due to its exceptional rendering speed and quality. Most initial approaches [26, 35, 57, 15, 63, 36, 70, 79, 68] relied on per-scene optimization routines and abundant multi-view images, which can limit broader applicability. Recently, researchers have begun to investigate feed-forward pipelines [10, 11] that, once trained, can generate novel views from sparse input images in a single inference step. However, to the best of our knowledge, there is currently no 3DGS framework specifically tailored for sparse-view 3D animal reconstruction.

## 3 Method

At a high level, *Pose Splatter* uses masked images captured from a small number of calibrated cameras. We apply shape carving techniques to create a voxelized "rough" representation of a single animal's pose and appearance. Then the rough volume passes through a stacked 3D U-Net architecture to produce a "clean" volume with an occupancy channel and additional feature channels. The occupancy channel determines whether to render a Gaussian for each voxel, while the remaining feature channels for each rendered voxel are independently mapped through a small MLP to determine 3D displacement vectors, which de-voxelizes the representation, along with covariance and appearance features needed to render the Gaussian. Finally, the scene is rendered given camera parameters by Gaussian splatting. Image-based losses are then used to propagate gradients through the model parameters. The model is a simple, lightweight framework which uses only about 2.5 GB of GPU memory (VRAM). Note that *Pose Splatter* captures an animal's pose at a given instant not in the sense that it estimates joint angles or positions, but in the more general sense of capturing the whole geometry of the animal. The overall framework is illustrated in Figure 1a. See Appendix A for a brief discussion of camera considerations.

**Mask Generation**  We generate mask videos used for training using a pre-trained Segment Anything Model (SAM2) [51]. We first prompted a mask on the first frame using SAM2 image mode, and propagate the mask through the video using SAM2 video mode. We chose not to fine-tune SAM2 on our video datasets to see how well our pipeline could perform without manual annotation, but it is also possible to fine-tune the model, which would likely improve the results.

**Determining Animal Position and Rotation**  The model quantifies animal pose independent of the 3D position and azimuthal orientation (about the vertical axis) of the animal. We determine these quantities without relying on 3D keypoints, thereby avoiding the time-intensive process of creating a training dataset for a keypoint detection network. To this end, a robust triangulation of the center coordinates of the mask in each image yields a rough 3D center. Shape carving, described below, provides a rough estimate of the animal mass, which is summarized as a 3D Gaussian distribution with a mean vector and covariance matrix. The mean vector is taken as the animal's position, while

the principal axis of the 3D covariance matrix is tracked smoothly over time and projected onto the X/Y plane to estimate the horizontal rotation. Additional details can be found in Appendix B.

**Shape Carving Procedure** Shape carving centers a 3D cubic grid of voxels at the estimated animal center and rotates it according to the estimated azimuthal rotation angle. By back-projecting masks from each camera view into space, we determine which voxels lie outside the visual hull of the animal and which ones are inside. We assign colors to the inside voxels based on their visibility from each view. This approach assumes minimal occlusion and relies on precise camera calibration to ensure accurate alignment of projections. In particular, this precludes a direct application to multi-animal recordings, where occlusions are common. Figure 1b provides an intuitive illustration of the shape carving process. See Appendix C for more details on our shape carving procedure.

**Stacked U-Net Architecture** The stacked 3D U-Net architecture consists of three U-Net modules arranged sequentially, designed to progressively refine the rough volume. This stacked configuration, inspired by the refinement capability of stacked hourglass networks, allows for iterative enhancement of feature quality and spatial coherence. Each module contains four downsampling and upsampling blocks, with skip connections to facilitate feature preservation. All layers employ ReLU activations, and the last U-Net module outputs 8 channels. The U-Nets are initialized to approximate the identity function by initializing filters near the Dirac delta filter and relying on the first skip connection to propagate the image through the U-Net. We find that no pretraining is needed with this initialization, unlike standard initialization schemes.

**Gaussian Splatting** To render the animal using Gaussian splatting, we first decide which Gaussians to render, interpreting the first channel of the volume as a probability of rendering a Gaussian for a given voxel. For each rendered voxel, we pass all 8 channels through a small MLP to produce Gaussian parameters. Crucially, the mean parameter is taken by adding the MLP-predicted displacement to the coordinate of the voxel in 3D space, thereby de-voxelizing the animal shape representation. A standard splatting is then performed, taking Gaussian and camera parameters and outputting a rendered RGBA image. We use the gsplat library [72] to perform splatting. In standard splatting, each Gaussian particle is described by a location parameter $\mu \in \mathbb{R}^3$ and a spatial covariance matrix $\Sigma \in \mathbb{R}^{3 \times 3}$, in addition to color and opacity parameters, $c \in \mathbb{R}^3$ and $\alpha \in [0, 1]$, respectively. The density of the Gaussian particle at location $x$ is given by

$$G(x) = \exp\left(-\tfrac{1}{2}(x - \mu)^\top \Sigma^{-1}(x - \mu)\right) .$$

As described in [84], using a camera transformation matrix $W$ and the Jacobian matrix $J$ of an affine approximation to the projective transformation, the resulting 2D covariance matrix is given by $\Sigma' = JW\Sigma W^\top J^\top$. Lastly, the color at a given pixel in the image plane is given by

$$\sum_{i=0}^{N} c_i \alpha_i \prod_{j=0}^{i-1}(1 - \alpha_j)$$

where the particles are ordered by decreasing depth (along the $j$ subscript) relative to the camera.

**Loss Terms** We employ two standard image-based losses to train the network. First, we calculate an L1 color loss to encourage accurate color rendering, as L1 losses are more robust to changes in colors caused by non-Lambertian effects than L2 losses: $\mathcal{L}_{color} = \sum_{ij} |\hat{x}_{ij} - x_{ij}| \,/\, 3\sum_{ij} m_{ij}$ where $\hat{x}$ is the predicted image, $x$ is the ground truth image with the rendering background color, outside of the masked region, modified to be white, and $m$ is the input mask. Second, we use an intersection-over-union loss to compare an input mask $m$ to the transparency channel of the rendered image $\hat{m}$ to encourage accurate silhouettes: $\mathcal{L}_{IoU} = 1 - \sum_{ij} (\hat{m}m)_{ij} / \sum_{ij} (\hat{m} + m - \hat{m}m)_{ij}$ , where products are elementwise. The total loss is $\mathcal{L} = \mathcal{L}_{IoU} + \lambda_{color}\mathcal{L}_{color}$ where $\lambda_{color}$ is a hyperparameter.

### 3.1 A Visual Embedding Technique

In addition to producing collections of Gaussian particles that capture the 3D geometry and appearance of the animal, it is often useful to distill these particles into a moderate-dimensional descriptor of pose and appearance for subsequent behavioral analyses. For example, 2D and 3D keypoint coordinates in an egocentric reference frame have greatly advanced the study of animal behavior [61, 27]. Designing a comparable descriptor directly from the Gaussian particle parameters is complicated by two factors. First, the number of particles can vary from frame to frame. Second, Gaussians that are completely inside the surface layer of the animal volume do not materially affect appearance. To bypass these issues, we propose a method that relies on rendered appearance of the particles rather than on the raw particle parameters while maintaining invariance to animal position and azimuthal rotation.

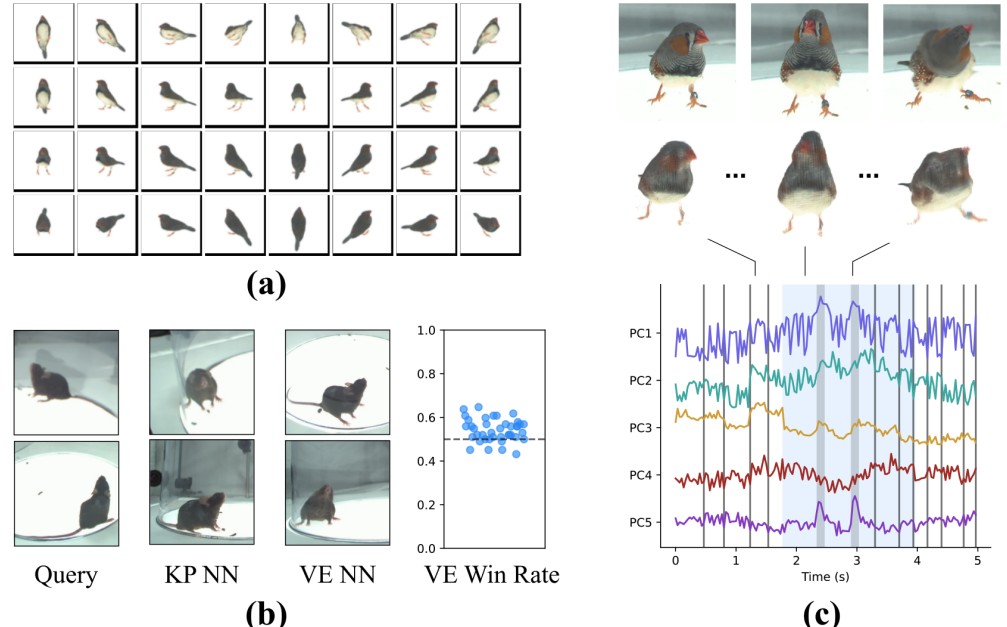

**(a)**

**(b)**

Query     KP NN     VE NN     VE Win Rate

**(c)**

Figure 2: **(a) Example renderings for visual embedding.** 32 virtual cameras, distributed on a sphere centered on the animal, produce appearance-only renderings used to build our visual embedding. **(b) Nearest-neighbor preference study.** Nearest-neighbor retrieval with the visual embedding (VE) is favored over a 3D-keypoint (KP) baseline in a 102-person study (54 % vs. 50 %; $p = 1.5 \times 10^{-5}$, two-sided $t$-test, $n = 40$). The query pose (left) and its two candidate nearest neighbors illustrate that the visual embedding preserves a subtle leftward head tilt that the keypoint method misses. **(c) Visual embedding tracks subtle movements.** Two image rows are shown—the upper row contains ground-truth frames and the lower row the corresponding *Pose Splatter* renders, illustrating from left to right a typical pose, slight feather expansion, and a head-shaking bout. Beneath them, the first five principal components (PC1–PC5) plotted through time reveal these behaviors: thin grey lines indicate head reorientations that coincide with changes in PCs 2–4, dark-grey bands mark brief head-shaking bouts that stand out in PCs 1 and 5, and the surrounding light-blue interval captures slow feather expansion and compression, reflected in the low-frequency trends of PCs 1 and 2.

**Overview of the Embedding** We place a virtual camera at a set of viewpoints covering the sphere centered on the animal's 3D center. At each viewpoint $(\theta, \phi)$, where $\theta \in [0, \pi]$ is the polar angle from the positive $z$-axis down and $\phi \in [0, 2\pi]$ is the azimuthal angle about the $z$-axis, we render a $224 \times 224$ RGB image of the Gaussian particles as seen looking inward toward the animal (Figure 2a). Because internal Gaussians are occluded in all views, the resulting set of images captures only the visually relevant features (shape, silhouette, color, etc.).

**Latent Encoding** Rather than working directly with the RGB images, each $224 \times 224$ rendering is passed through a pretrained convolutional encoder (ResNet-18) that outputs a 512-dimensional feature vector. Denote this resulting function on the sphere by $f(\theta, \phi) \in \mathbb{R}^{512}$. Each component $f_k(\theta, \phi)$ (for $k = 1, \ldots, 512$) encodes some learned feature of appearance or geometry across the sphere.

**Spherical Harmonic Expansion and Quadrature** To produce a rotation-invariant descriptor, we expand each component $f_k$ in a truncated spherical harmonic basis $Y_{\ell m}(\theta, \phi)$ of bandwidth $L$,

$$f_k(\theta, \phi) \approx \sum_{\ell=0}^{L} \sum_{m=-\ell}^{\ell} \hat{f}_{k,\ell m} \, Y_{\ell m}(\theta, \phi).$$

We estimate each coefficient $\hat{f}_{k,\ell m}$ via spherical quadrature:

$$\hat{f}_{k,\ell m} \approx \sum_{j=1}^{N_\theta} \sum_{i=1}^{N_\phi} w_{j,i} \, f_k(\theta_j, \phi_i) \, Y_{\ell m}^*(\theta_j, \phi_i),$$

where $(\theta_j, \phi_i)$ range over a suitable sampling grid on the sphere, and $w_{j,i}$ are the corresponding quadrature weights. We use Gauss-Legendre quadrature with $L = 3$, which avoids evaluating the points $\theta \in \{0, \pi\}$, where there is no well-defined vertical camera orientation.

**Ensuring Invariance to Horizontal Rotations** Our goal is invariance to rotations of the animal about the vertical ($z$) axis—i.e. a shift in $\phi$. Under such a rotation, $\hat{f}_{k,\ell m}$ picks up a phase factor $e^{i\,m\,\phi}$. By taking the squared magnitude $\|\hat{f}_{k,\ell m}\|^2$, we eliminate that phase dependence. Consequently, we form our final descriptor by collecting all such terms for each latent dimension $k$ and each $\ell, m$:

$$\left\{ \|\hat{f}_{k,\ell m}\|^2 \right\}_{k=1..512,\ \ell=0..L,\ m=-\ell..\ell}.$$

This yields a fixed-size feature vector whose entries remain the same if the animal is rotated in the horizontal plane.

However, we found that these feature vectors still strongly encoded the azimuthal angle of the animal, possibly due to uneven lighting conditions across views. To remove this effect, we employ an adversarial formulation of principal components analysis (PCA) to find a 50-dimensional subspace of the feature vectors that contains a large portion of the variance of the input vectors and can predict only a small portion of the variance of the sine and cosine components of the azimuthal rotation angle [9]. We take these 50-dimensional pose descriptors as the visual embedding. See Appendix D for more details.

## 4 Experiments

### 4.1 Datasets

Our first dataset consists of six synchronous 30-minute videos taken from cameras with known parameters of a freely-moving mouse in a 28 cm diameter plastic cylinder, which is released as a publicly available dataset accompanying this paper [17] (CC0 1.0). The RGB videos are captured at a resolution of $1536 \times 2048$ at a frame rate of 30 FPS, for a total of 324000 frames. In the video, the mouse engages in a variety of behaviors including walking, rearing, grooming, and resting. Consecutive thirds of the video are used for training, validation, and testing. A second dataset of a freely-moving zebra finch is obtained in the same manner with a 20 minute duration. Both datasets are downsampled spatially by a factor of 4 and temporally by a factor of 5 in the following results.

A third dataset is Rat7M, which contains videos of a freely moving rat in a cylindrical arena captured from 6 camera angles (CC BY 4.0) [39]. This video is more challenging to mask due to occlusions of the feet and tail by the bedding material, additional occlusion on the side of the arena in one of the views, and uneven lighting conditions across the views. We present results on a subset of 135,000 frames. Appendix E contains additional results from a subset of the 3D-POP dataset consisting of a single freely moving pigeon in a large room captured from 4 camera angles (CC BY 4.0) [44, 43].

### 4.2 Training Details

The loss hyperparameter was tuned by hand to encourage realistic renderings on the training set of the first mouse video. We set $\lambda_{color} = 0.5$ for all experiments. We trained *Pose Splatter* with a single Nvidia RTX A4000 GPU along with 32 CPU cores used for data fetching. Our model uses only 2.5 GB of GPU memory (VRAM), compared to 10GB for Gaussian Object [68] and 20GB for PixelSplat and MVSplat [10, 11], thanks to its simple architecture. Training runs varied between 2 and 12 hours, depending on the number of frames and camera views used. The learning rate was fixed at $10^{-4}$ for all experiments. The number of epochs was chosen so to minimize validation set loss, and ranged from 40 to 75 epochs across all experiments. A single forward pass through the model takes about 30 ms during inference. Full details are in Appendix F. Project code is available at `https://github.com/jackgoffinet/pose-splatter`.

### 4.3 Rendering Metrics

We report four metrics to compare the quality of rendered images to ground truth. First, intersection over union (IoU) computes the ratio of the intersection of the binarized predicted mask and the ground truth mask to their union, measuring how well the predicted geometry aligns with the true object silhouette. Second, the average L1 distance between predicted and ground truth colors, normalized by the ground truth mask area, quantifies the accuracy of surface appearance. Third, the peak signal-to-noise ratio (PSNR) evaluates the overall pixel-wise fidelity of the rendered image, with higher values indicating lower reconstruction error relative to the dynamic range of image intensities.

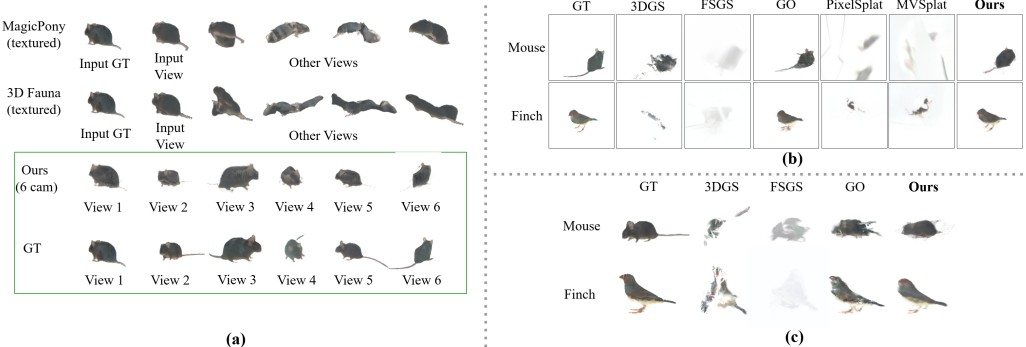

Figure 3: **(a) Single-view reconstruction.** Against single-view baselines, *MagicPony* and *3D Fauna* collapse when the camera departs from the input view, failing to recover a plausible mouse geometry. *Pose Splatter*, by contrast, reconstructs accurate shapes from all viewpoints of an unseen time step in the test set. **(b) Sparse-view 3DGS comparison.** Most sparse-view 3DGS baselines reproduce the white background well but fail to reconstruct the given subject. Consequently, their quantitative scores appear high even though the rendered animals lack detail. See Table 1a for quantitative scores. **(c) Comparison with per-scene-optimized 3DGS (4 view).** Some methods post good metrics yet still fail to reconstruct the given subject. See Table 1b for metrics. Only GaussianObject and *Pose Splatter* deliver comparable, visually convincing foreground reconstructions.

Lastly, the structural similarity index measure (SSIM) assesses the perceptual similarity between the predicted and ground truth images by comparing local luminance, contrast, and structural information, providing a more perceptually aligned quality score than pixel-wise metrics.

## 4.4 Results

*Pose Splatter* accurately learns animal geometry and appearance. We compared *Pose Splatter* with the strongest sparse-view 3D Gaussian–splatting (3DGS) baselines. Because the highest-performing methods to date still rely on scene-specific optimization, we first selected three per-scene optimization pipelines: 3DGS [26], FSGS [79], and GaussianObject (GO) [68]. For every test scene, each model was optimized from scratch on the same five input views, leaving the remaining single view unseen for evaluation, and we retained the authors' default hyperparameters throughout. As summarized in Table 1a and illustrated in Figure 3b, *Pose Splatter* outperforms every baseline on both evaluation datasets. The

| Method | | Mouse | | | | Finch | | | |
|---|---|---|---|---|---|---|---|---|---|
| | | IoU↑ | L1↓ | PSNR↑ | SSIM↑ | IoU↑ | L1↓ | PSNR↑ | SSIM↑ |
| *Per-Scene Optimization* | 3DGS | 0.502 | 0.742 | 25.9 | 0.969 | 0.513 | 0.689 | 26.4 | 0.975 |
| | FSGS | 0.462 | 0.923 | 25.3 | 0.975 | 0.454 | 0.925 | 25.6 | 0.981 |
| | GO | 0.732 | **0.628** | 28.8 | 0.977 | 0.819 | 0.382 | 34.1 | 0.990 |
| *Feed-Forward* | PixelSplat | 0.424 | 0.921 | 25.2 | 0.968 | 0.428 | 0.858 | 26.2 | 0.971 |
| | MVSplat | 0.417 | 0.887 | 25.5 | 0.966 | 0.461 | 0.893 | 25.9 | 0.970 |
| | **Ours** | **0.760** | 0.632 | **29.0** | **0.982** | **0.848** | **0.345** | **34.5** | **0.992** |

(a) Comparison with sparse-view 3DGS methods.

| Method | Mouse (4 cam) | | | | Finch (4 cam) | | | |
|---|---|---|---|---|---|---|---|---|
| | IoU↑ | L1↓ | PSNR↑ | SSIM↑ | IoU↑ | L1↓ | PSNR↑ | SSIM↑ |
| 3DGS | 0.447 | 0.786 | 25.8 | 0.967 | 0.459 | 0.754 | 26.1 | 0.973 |
| FSGS | 0.414 | 0.982 | 24.9 | 0.974 | 0.423 | 0.891 | 25.4 | 0.980 |
| GO | 0.706 | **0.745** | **28.5** | 0.981 | 0.725 | **0.657** | **30.4** | **0.985** |
| **Ours** | **0.721** | 0.753 | 28.2 | **0.982** | **0.731** | 0.685 | 29.0 | 0.981 |

(b) Comparison with per-scene optimization methods.

Table 1: **(a) Sparse-view 3DGS benchmark.** We compare our method with three per-scene optimization 3DGS baselines and two feed-forward alternatives. Higher values for IoU, PSNR, and SSIM and lower values for L1 indicate better performance. The best score in each column is set in bold; the second-best is underlined. See Figure 3b for qualitative results. **(b) Additional benchmark.** We conducted additional evaluation by limiting the input to just 4 views and benchmarking against the optimization-based 3DGS baselines. See Figure 3c for qualitative results.

original 3DGS, which is designed for dense multi-view input, degrades noticeably when given only sparse cameras (see Figure 3b); its scores remain moderate largely because it reproduces the uniform white background well, which inflates pixel-wise similarity even as the animal's geometry collapses.

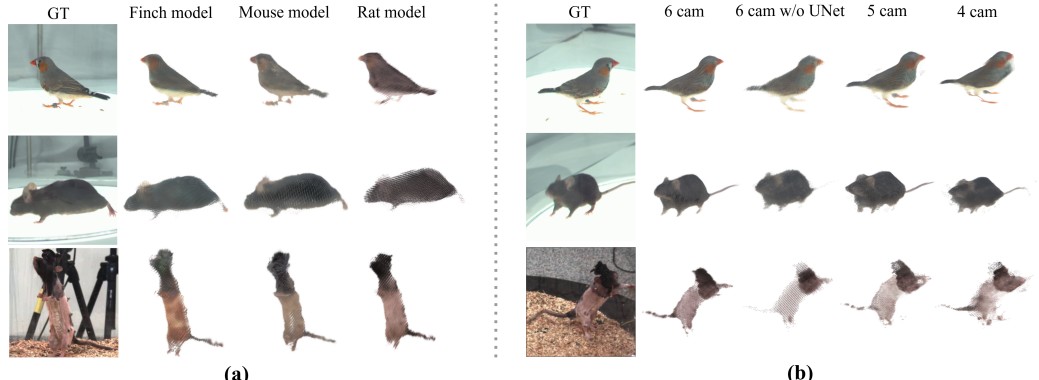

GT    Finch model    Mouse model    Rat model        GT    6 cam    6 cam w/o UNet    5 cam    4 cam

**(a)**                              **(b)**

Figure 4: **(a)** Cross-species renderings. **(b)** Renderings given different numbers of input views. The rendered views are novel for the 5- and 4-camera models.

FSGS behaves much the same: it reproduces the white background well and sketches a coarse shape, so its overall metrics stay reasonable, yet—as Figure 3b shows—it fails to recover the correct body shape. GaussianObject (GO) achieves the strongest numbers among the optimization methods. By coupling pretrained diffusion priors with iterative depth-guided refinements, GO produces markedly cleaner surfaces and sharper texture, as both the table and figure confirm. The trade-off is runtime: each scene of GO requires roughly one hour of test-time-optimization, as opposed to the roughly 30 ms test-time forward pass of *Pose Splatter* (over 100,000x faster).

We now turn to feed-forward baselines. In line with each author's recommendations, we trained both PixelSplat and MVSplat with two input views and treated one of the remaining views as the test view, leaving all other hyperparameters at their defaults. Both models post moderate scores because, like the other methods, they reproduce the white background. However, Figure 3b shows that neither network reconstructs the animal itself with meaningful accuracy. Both authors note that their pipelines depend on substantial overlap between input views to establish reliable cross-view correspondences. Our datasets provide minimal overlap, leaving few shared features to match, and this limitation prevents either model from capturing the true 3D geometry. We experimented with varying the number of input views during testing, but observed no noticeable changes in the results (see Appendix G). In addition, we provide comparisons with several large, pretrained, and generalizable feed-forward 3D reconstruction models in Appendix G, including HunYuan 3D-2 [78], TRELLIS [65], VGGT [60], and AnySplat [23].

We further evaluated the optimization-based baselines and *Pose Splatter* using a sparser view setting, with each method trained on only four views and assessed on the remaining two. Results are in Table 1b and Figure 3c. Under this split, GaussianObject retains a slight edge, but *Pose Splatter* matches it within a small margin on every metric. Figure 3c shows that both methods capture comparable geometry and texture, whereas the original 3DGS and FSGS do not perform well. 3DGS and FSGS achieve good numerical scores because they accurately reconstructed the white background, but qualitative experiments make clear that fine detail is lost.

Additionally, because multi-view animal datasets are scarce, the current state of the art in 3D animal reconstruction still relies on single-image mesh predictors. We therefore compared *Pose Splatter* with two leading single-view models: MagicPony [64] and 3D Fauna [33]. We trained these two models on all six reference views and evaluated them on a random view from an unseen time-step. *Pose Splatter* used the same six training images but was tested on all six views of the unseen time step. As illustrated in Figure 3a, the single-view networks accurately reproduce their input view yet fail to maintain shape coherence once the mesh is rotated, whereas *Pose Splatter* preserves plausible anatomy from every angle. The difference stems from the fact that single-image pipelines never observe the six views simultaneously, making it difficult to resolve self-occlusions and silhouette ambiguities in complex animal poses. We also present the results of another single-view animal reconstruction model, BANMo [69], on our dataset in Appendix G. Similar to MagicPony and 3D Fauna, BANMo fails to accurately reconstruct the 3D shape from unseen views. See Appendix H for additional novel and input-view *Pose Splatter* renderings.

| Method | Mouse | | | | Finch | | | | Rat | | | |
|---|---|---|---|---|---|---|---|---|---|---|---|---|
| | IoU↑ | L1↓ | PSNR↑ | SSIM↑ | IoU↑ | L1↓ | PSNR↑ | SSIM↑ | IoU↑ | L1↓ | PSNR↑ | SSIM↑ |
| 6 cam | **0.868** | **0.317** | **33.5** | **0.989** | **0.913** | **0.231** | **36.4** | **0.991** | **0.797** | **0.658** | **26.9** | **0.975** |
| 6 cam⁻ | 0.825 | 0.380 | 32.2 | 0.987 | 0.876 | 0.308 | 34.5 | 0.990 | 0.664 | 0.849 | 25.5 | 0.971 |
| 5 cam | **0.760** | **0.632** | **29.0** | 0.982 | **0.848** | **0.345** | **34.5** | **0.992** | **0.794** | **0.628** | **27.6** | **0.981** |
| 5 cam⁻ | 0.748 | 0.663 | 28.8 | **0.983** | 0.838 | 0.421 | 33.7 | 0.991 | 0.688 | 1.16 | 24.6 | 0.970 |
| 4 cam | **0.721** | **0.753** | 28.2 | **0.982** | **0.731** | **0.685** | **29.0** | **0.981** | **0.651** | 1.16 | **24.4** | **0.967** |
| 4 cam⁻ | 0.701 | 0.737 | **28.4** | 0.982 | 0.675 | 0.874 | 28.0 | 0.979 | 0.579 | 2.01 | 23.5 | 0.955 |

(a) *Pose Splatter* ablation study

| | IoU↑ | L1↓ | PSNR↑ | SSIM↑ |
|---|---|---|---|---|
| Mouse → Rat | **0.658** | **1.014** | **25.1** | **0.972** |
| Finch → Rat | 0.545 | 1.200 | 24.0 | **0.972** |
| Mouse → Finch | 0.719 | 0.625 | 31.1 | 0.988 |
| Finch → Mouse | 0.736 | 0.609 | 29.3 | 0.982 |

(b) 5-camera cross-species generalization

Table 2: **(a) Ablation study.** The table shows how *Pose Splatter* responds to fewer input views and to the removal of the stacked U-Net refinement (methods annotated with a superscript minus). **(b) 5-camera cross-species generalization.** Models trained on one species are evaluated on the single held-out view of another, revealing how well the learned representation transfers across animals.

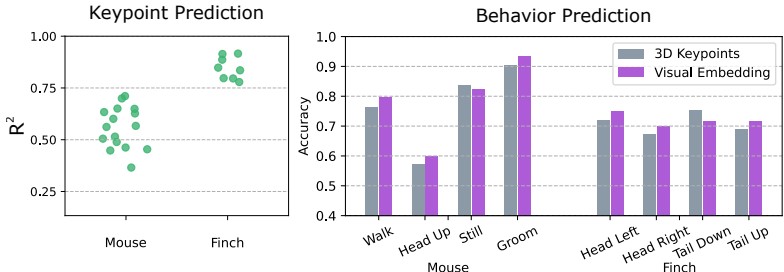

Figure 5: **Left** $R^2$ values of predicting egocentric 3D keypoints from visual embeddings. Each scatterpoint represents a single manually annotated keypoint. **Right:** Accuracies of logistic regression models predicting different manually annotated behaviors using egocentric 3D keypoints (gray) versus visual embeddings (purple). Six of eight behaviors are better predicted by the visual embedding.

*Pose Splatter* **generalizes across different species** We evaluated cross-species transfer by applying models trained on one animal to a single held-out view of another. As Table 2b shows, accuracy trails the in-species baselines (cf. the 5 camera results in Table 2a), yet the decline is modest; indeed, the Finch-to-Mouse model matches the Mouse-only 5-camera model. Also, the stronger Mouse-to-Rat performance, compared with Finch-to-Rat, suggests that morphological similarity may ease transfer. Qualitative results in Figure 4a confirm that, despite the domain shift, all three models still recover the novel animal's overall shape and appearance with only minor degradation.

**Visual embedding produces preferred nearest neighbors** We have seen that *Pose Splatter* is able to accurately model the pose and appearance of animals. Now we turn our attention to whether the proposed visual embeddings provide an informative description of animal pose. First, we reasoned that a good description of animal pose should provide meaningful nearest neighbors. To test this, we trained a supervised 2D keypoint predictor (SLEAP, [48]) using 1000 hand-labeled images of mouse poses with 16 keypoints, which required roughly 15 hours of manual annotation time. We then performed a robust triangulation to create 3D keypoints for each frame. The keypoints are shifted and rotated in the X/Y plane into an egocentric coordinate system, a 48-dimensional representation of mouse pose. We additionally calculated our proposed visual embedding, a 50-dimensional representation, which requires no manual annotation phase. To gauge the relative quality of nearest neighbors produced by both feature sets, we had 102 participants chose the more similar of two poses to 40 randomly sampled query poses. We chose a Euclidean metric to calculate nearest neighbors in both feature spaces and excluded nearby frames in time (within 500 frames). We additionally randomized the order of presentation of the two candidate answers. Example queries and answers are shown in Figure 2b, demonstrating the high degree of similarity produced by both feature sets. Participants showed a slight but statistically significant preference for the visual embedding nearest neighbors ($54.0 \pm 0.8\%$ of visual embedding nearest neighbors preferred, mean $\pm$ SEM, $p = 1.5 \times 10^{-5}$, two-sided one-sample $t$-test, $n = 40$). Thus, our findings provide strong evidence of a genuine preference for the visual embedding neighbors over the keypoint-based neighbors. This is despite the fact that *Pose Splatter* requires **no manual annotations** and the 3D keypoints were tested in the optimistic condition where the manual annotations were taken from the same dataset. See

Appendix I for an investigation of the triangulated 3D keypoint quality and Appendix J for additional survey details.

**Visual embedding captures subtle movements**  To test whether the visual embedding could encode subtle variations in pose, we took a closer look at a 5-second clip in the test portion of the finch video in which the finch's feathers slowly but subtly expand in a way not seen in the training footage. Interspersed in this footage are two brief bouts of head shaking. Figure 2c shows three stills from the clip in addition to the same three rendered frames. Below, we plot the first 5 principal components of the visual embedding over time and observe that they correspond very well with the annotated features of the video, including the subtle expansion of feathers.

**Visual embedding vs. keypoints**  We next assessed whether the visual embedding implicitly encodes 3D information by predicting egocentric 3D keypoints from the visual embedding using a 5-nearest-neighbor regressor (see Appendix D). The visual embedding explains the majority of variance for most mouse keypoints and all finch keypoints, even though it was not trained explicitly for this purpose (Fig. 5, left). We then evaluated a common downstream application—behavior classification—by training logistic regression models to predict manually annotated behaviors from either egocentric 3D keypoints or visual embeddings (see Appendix D for details). The visual embedding outperforms the keypoint-based features for six of the eight annotated behaviors, despite requiring no manual supervision (Fig. 5, right).

**Ablation study**  We measured how *Pose Splatter* performs when fewer input cameras are available and when the stacked U-Net refinement is removed. In the "shape-carving only" variant – which bypasses the U-Net and maps the carved voxel grid directly to Gaussian parameters – we flag each row with a superscript minus in Table 2a (e.g., 6 cam⁻). We observe that quantitative scores decline as views are removed yet the 5- and 4-camera models still recover the animals' overall shape and appearance well as shown in Figure 4b. Also, every setting that retains the stacked U-Net outperforms its counterpart without it. Qualitative results in Figure 4b clearly show this trend: compared with the full 6-camera model, the 6 cam⁻ ("6 cam w/o U-Net") volume is noticeably noisier, leading to blurrier renders and a loss of fine detail.

## 5  Discussion & Conclusion

In this work, we introduced *Pose Splatter*, a novel approach leveraging 3D Gaussian splatting and shape carving to reconstruct and quantify the 3D pose and appearance of laboratory animals. Unlike existing keypoint- and mesh-based methods, which either require extensive manual annotations or rely on predefined template models, *Pose Splatter* operates in a feed-forward manner without the need for per-frame optimization and requires no manual annotation. Our method successfully captures fine-grained postural details and provides a compact, rotation-invariant visual embedding that can be seamlessly integrated into downstream behavioral analyses.

Although *Pose Splatter* performs well with a single subject, some neuroscience experiments involve multiple interacting animals. Such settings introduce prolonged and severe occlusions beyond our current benchmarks. While *Pose Splatter* effectively resolves most self-occlusions, addressing these more challenging scenarios remains an open problem for future work. Additionally, while our visual embedding provides a powerful descriptor for capturing pose and appearance, further exploration is needed to improve interpretability and facilitate direct comparisons across species.

## Acknowledgments

We thank Tim Dunn, Kafui Dzirasa, and Kathryn Walder-Christensen for helpful conversations, which greatly improved the work. We are grateful to Katherine Kaplan and Rich Mooney for help obtaining zebra finch recordings and to Kathryn Walder-Christensen and Hannah Soliman for help obtaining mouse recordings. Lastly, we thank Kyle Severson for assistance with video acquisition software. Research reported in this publication was supported by the National Institute Of Mental Health of the National Institutes of Health under Award Number R01MH125430. The content is solely the responsibility of the authors and does not necessarily represent the official views of the National Institutes of Health.

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

# A  Camera Considerations

*Pose Splatter* requires a minimum of roughly four calibrated cameras to operate, but performs best with 5 or more cameras, as seen in Figure 4b. Additionally, even lighting conditions and well-spread camera orientations facilitate rotation-invariant visual embeddings and higher quality shape-carved volumes, respectively. These camera requirements may hinder application in some outdoor or wild settings, but are becoming more common in laboratory settings where precise behavioral tracking is needed (e.g. [2, 12, 61]).

# B  Center and Rotation Estimation

To estimate the animal's orientation at each time point $t$, we modeled its shape as a Gaussian distribution $\mathcal{N}(\boldsymbol{\mu}, \Sigma)$ by matching the 3D moments of a shape-carved voxel grid. We then extracted the principal axis by computing the eigenvector corresponding to the largest eigenvalue of $\Sigma_t$. This eigenvector is normalized to unit length and called $\mathbf{v}_t$. Because an eigenvector can flip sign from one time point to the next, we introduced a sign-consistency procedure to best ensure a smooth progression of the eigenvectors. Specifically, after computing $\mathbf{v}_{t+1}$ at time $t+1$, we used a Wasserstein-2 optimal transport map to track a reference point from the distribution at time $t$ to that at time $t + 1$. We then compared the distance of this transported point to the two possible orientations ($\boldsymbol{\mu}_{t+1} \pm \mathbf{v}_{t+1}$) and chose the orientation that preserved the local consistency (i.e., whichever was closer to the transported point).

Finally, we enforced a global orientation consistency by making use of the following heuristic: animals move on average in the direction they face, and not the opposite direction. We operationalize this heuristic by checking if the cumulative displacement of the mean positions $\boldsymbol{\mu}_t$ from start to end correlated positively with the sequence of principal axes $\{\mathbf{v}_t\}$. If the overall dot product was negative, we flipped all axes to align with the general direction of motion. This procedure yielded a temporally consistent set of principal axes $\{\mathbf{v}_t\}$ that reliably tracked the animal's primary orientation over time.

# C  Shape Carving Details

**Voxel Occupancy**  We first start with a $112 \times 112 \times 112$ spatial grid with equal-sized edges. The $z$-axis is aligned with the third axis of the grid, while the first axis of the grid is aligned with the estimated azimuthal heading direction of the animal (see Appendix B). For each point in the grid and each camera, we determine whether its projection onto the image plane of the camera corresponds with a masked (animal) or unmasked (background) point. Voxels that correspond with masked regions in at least $N$ cameras are considered occupied.

**Voxel Colors**  To estimate the color of each voxel in the reconstructed volume, we first determined its visibility from multiple camera viewpoints using a ray-casting procedure. For each camera, the voxels were sorted by their distance from the camera center, ensuring that the closest voxels were processed first. These voxels were then projected into the image plane using a standard pinhole camera model. To track occlusions, a depth buffer was maintained at each pixel location. If a voxel mapped to a pixel that already contained a closer voxel, it was marked as occluded. This process produced a visibility mask indicating whether each voxel was directly observable from each camera.

We then computed each voxel's color by sampling the corresponding pixel values from the camera images. All voxels, whether fully visible or occluded, were projected into the camera views, and their colors were retrieved from the respective images. To integrate these sampled colors into a single representative value per voxel, we applied a weighted averaging scheme. Fully visible voxels contributed with a weight of one, while occluded voxels were assigned a reduced weight (0.25) to reflect the uncertainty introduced by occlusion. The final voxel color was obtained by normalizing and summing these weighted contributions across all cameras, allowing us to account for occlusions while still incorporating partial information from less reliable viewpoints.

**Determining Volume Dimensions**

To save on RAM and GPU memory during training, we truncated our original $112 \times 112 \times 112$ volumes based on voxel usage in the animal's training frames. We first calculated a sum over all the voxel occupancies and then manually set thresholds and determined volume slice indices to balance

voxel use and memory considerations. The following table shows the final dimensions of the volumes for each animal.

| Mouse | $96 \times 80 \times 64$ |
|-------|--------------------------|
| Finch | $96 \times 64 \times 80$ |
| Rat   | $96 \times 80 \times 64$ |

Table 3: Volume dimensions ($d_x \times d_y \times d_z$) for each animal.

**Final Steps** Our volumes consist of 4 channels: one binary occupancy channel and 3 color channels. To produce our final volumes, we produce two volumes independently, one with an occupancy threshold of $C$, where $C$ is the number of cameras, and one with the threshold $C - 1$. We then average the two volumes together. Note that the $C - 1$ threshold produces a coarser visual hull, with generally more occupied voxels. Initial tests showed that fine body parts such as mouse tails were often not represented in the shape-carved volume with just a threshold at $C$, especially when masks from different views disagree on the boundaries of the animal. We found that adding the $C - 1$ threshold and averaging resulted in much better coverage of fine body parts.

## D   Visual Embedding Details

The feature extractor used in the visual embedding is a pre-trained ResNet 18, trained on the ImageNet dataset. We pass rendered images through all but the last layer of the network, producing a 512-dimensional vector per image. 32 of these vectors are used in a spherical routine to produce the norms of 16 spherical harmonic coefficients, independently for each feature dimension. The resulting coefficients are then flattened to 8192-dimensional vectors.

To produce our visual embeddings, we reduce the feature dimension in two steps. First, we perform PCA on dataset of feature vectors, reducing the feature vectors to 2000 dimension. Then we collect the estimated azimuthal angle for each pose and concatenate the sine and cosine components of this angle together to be used as concomitant data in the adversarial PCA routine [9]. Adversarial PCA was then performed to reduce the feature dimensionality to 50 from 2000. The regularization strength parameter $\mu$ was set to the smallest integer power of 10 such that the adversarial PCA reconstructions of the sine and cosine components produced an average $R^2$ value less than 0.05, indicating the sines and cosines are not readily linearly decodable from the 50-dimensional features. We call these features the *visual embedding*.

To predict egocentric 3D keypoints from visual embeddings (Figure 5, left), we used a consecutive 80/20 train/test split, fit a 5-nearest neighbor regressor to the training data, and report a uniform average of the $R^2$ scores over the 3 spatial dimensions on the test set. We also predicted egocentric 3D keypoints from the 8192-dimensional visual features, which are processed to produce the 50-dimensional visual embeddings to test how much usable information is lost in this process. We used ridge regression with 5-fold cross validation on the training set to select a model and then report a uniform average of the $R^2$ scores over the 3 spatial dimensions on the test set. Figure 6 shows moderately better performance using the visual features than the visual embedding to predict egocentric 3D keypoints.

To classify behavior given egocentric 3D keypoints and visual embeddings, we used a random 60/40 train/test split using all available frames. We used 5-fold cross validation to determine the L2 regularization strength on the train set, targeting a balanced accuracy metric. Accuracies for each class are then reported on the test set. Mouse behavior was classified using one 4-way prediction (Walk vs. Head Up vs. Still vs. Groom) and finch behavior was classified using two two-way predictions (Head Left vs. Head Right and Tail Down vs. Tail Up).

## E   Application to 4-View Pigeon Data

To complement the 6-view mouse, finch, and Rat7M datasets, we applied *Pose Splatter* to a subset (Sequence 8) of the 4-view 3D-POP dataset ([44]), which contains video of a single pigeon in a large room.

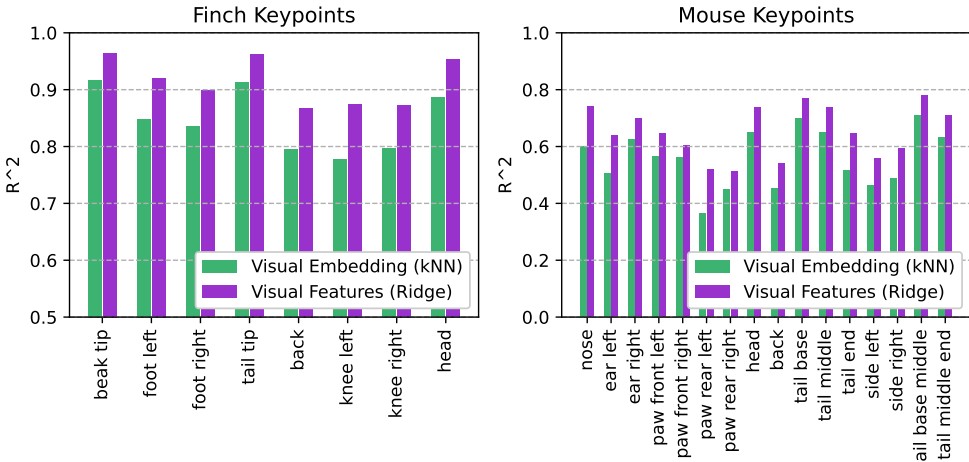

Figure 6: **Predicting egocentric 3D keypoints from visual embedding ($d = 50$) and visual features ($d = 8192$).** We observe good predictive ability as measured by $R^2$ values using both visual embeddings (green) and visual features (purple) to predict held-out finch (left) and mouse (right) keypoints. Note the different vertical axes in the two subplots.

Compared to the three datasets tested previously (mouse, finch, and Rat7M), the cameras in the 3D-POP data are located further apart relative to the size of the animal. For this reason, we found that the shape carving procedure with the provided camera parameters sometimes failed to detect overlap among the back-projected masks. To correct for this, we applied an adaptive frame-by-frame adjustment to the intrinsic camera parameters.

Briefly, we first find 2D mask centroids in each view and triangulate a rough 3D center of the animal using these centers and the provided camera parameters. We then re-project this 3D center point onto the 2D image planes and calculate a discrepancy between the reprojected point and the 2D mask centroids. Lastly, the camera center parameters ($c_x$ and $c_y$) are updated to remove the discrepancy.

More specifically, we assume an intrinsic matrix of the form

$$\begin{bmatrix} f_x & 0 & c_x \\ 0 & f_y & c_y \\ 0 & 0 & 1 \end{bmatrix} . \tag{1}$$

We then project the 3D center point $x_{\mathrm{world}}$ into camera coordinates: $x_{\mathrm{cam}} = R x_{\mathrm{world}} + t$, where $[R; t]$ are the camera's extrinsic parameters. Lastly, we update the intrinsic center parameters for each camera:

$$c_x \leftarrow u^* - f_x \big(x_{\mathrm{cam}}^{(1)}/x_{\mathrm{cam}}^{(3)}\big) , \qquad c_y \leftarrow v^* - f_y \big(x_{\mathrm{cam}}^{(2)}/x_{\mathrm{cam}}^{(3)}\big) \tag{2}$$

where $u*$ and $v*$ are the image coordinates of the reprojected 3D center and $\big(x_{\mathrm{cam}}^{(1)}, x_{\mathrm{cam}}^{(2)}, x_{\mathrm{cam}}^{(3)}\big)$ are the three coordinates of the 3D center in camera coordinates.

| IoU↑ | L1↓ | PSNR↑ | SSIM ↑ |
|---|---|---|---|
| 0.622 | 1.16 | 24.8 | 0.982 |

Table 4: *Pose Splatter* metrics on sequence 8 from the 3D-POP single pigeon, 4-view dataset.

Apart from this intrinsic parameter modification all aspects of the pipeline remain unchanged for the pigeon data. Table 4 shows performance metrics on the test set (c.f. Table 2a, 4 cameras). Qualitatively, the renderings are of lower quality than the mouse, finch, and rat 4-camera renderings, as expected from the lower PSNR, but still adhere to the overall shape of the pigeon (Figure 7).

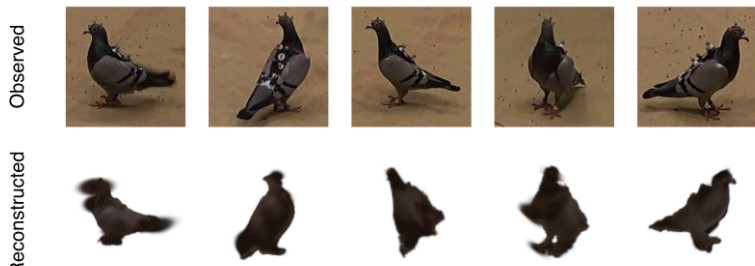

Figure 7: Representative test set pigeon reconstructions from sequence 8 of the 3D-POP dataset.

| | Shape Carving | U-Nets | MLP | Shift & Rotate | Splatting | Train FPS | Inference FPS |
|---|---|---|---|---|---|---|---|
| **Finch** | 7.0 ms | 4.6 ms | 14.3 ms | 6.0 ms | 0.2 ms | 7.0 | 7.4 |
| **Mouse** | 7.0 ms | 4.7 ms | 13.6 ms | 3.5 ms | 0.3 ms | 5.8 | 6.5 |
| **Rat** | 7.1 ms | 5.1 ms | 14.3 ms | 2.6 ms | 0.3 ms | 9.7 | 13.1 |

Table 5: Median times for the 5 stages of the *Pose Splatter* forward pass and overall frame rates for training and inference.

## F   Training and Network Details

Training was run with a single Nvidia RTX A4000 GPU along with 32 CPU cores used for data fetching. Our model uses only about 2.5GB of GPU memory (VRAM), thanks to its simple architecture. Training runs varied between 2 and 12 hours, depending on the length of the dataset. We estimate that the experiments presented in this paper required 6 days of compute time to complete.

The learning rate was fixed at $10^{-4}$ for all experiments. The number of epochs was chosen to minimize validation set loss, and ranged from 40 to 75 epochs across all experiments.

In experiments with fewer than 6 camera views, we found that this center and rotation estimation procedure produces less reliable results. While it would be useful to develop a more robust procedure for camera systems with fewer cameras, this was not a primary aim of our work. Therefore, we used the centers and orientations inferred with all 6 cameras for all experiments.

Tables 5 and 6 show timing information and the number of Gaussians rendered for all three datasets. Additional details may be found in the supplemental code.

Meanwhile, our additional training details for the comparison baselines were as follows. For the per-scene optimization methods (3DGS [26], FSGS [79], and GaussianObject [68]), the original pipelines initialize optimization with a point cloud created from "Structure-from-Motion Revisited (COLMAP)." In typical neuroscience-oriented animal-behavior studies, however, one may work with hundreds of thousands of video frames; running COLMAP on every frame would be prohibitively slow. We therefore assume no COLMAP ground-truth point cloud is available. Instead, we follow GaussianObject's COLMAP-free protocol: we seed optimization with a point cloud created from "DUSt3R: Geometric 3D Vision Made Easy (DUSt3R)," feeding the calibrated poses directly into DUSt3R (because the camera intrinsics and extrinsics are known) so that its reconstruction is as clean as possible. The GaussianObject paper shows that this DUSt3R initialization maintains state-of-the-art performance, a result that our experiments confirm. All remaining hyperparameters mirror the "best" settings recommended in each method's official repository. Additionally, per-scene optimization methods require varying amounts of time depending on the approach, ranging from a few minutes (3DGS, FSGS) to about an hour (GaussianObject) per scene. As such, optimizing tens of thousands of frames is impractical. Therefore, we randomly sampled 150 images from the test set and conducted experiments using this subset. For the feed-forward baselines, PixelSplat [10] and MVSplat [11], we likewise adopt the authors' recommended hyperparameters. Finally, the single-view animal reconstruction models MagicPony [64] and 3D Fauna [33] are trained with their prescribed data preprocessing pipelines and hyperparameters.

|        | Before Training | After Training |
|--------|-----------------|----------------|
| **Finch** | 13.9k | 13.8k |
| **Mouse** | 8.4k | 8.5k |
| **Rat** | 4.8k | 4.8k |

Table 6: Average number of Gaussians rendered before training (at initialization) and after training.

# G   Additional Model Comparisons

## G.1   PixelSplat & MVSplat

We report how the feed-forward baselines behave as the number of context views changes (see Figure 8). According to the papers and authors' code repositories, both PixelSplat [10] and MVSplat [11] were trained with two input views, and adding more cameras does not guarantee improvement in the output. Our results confirm that increasing the number of context views does not appreciably improve the performance of either baseline.

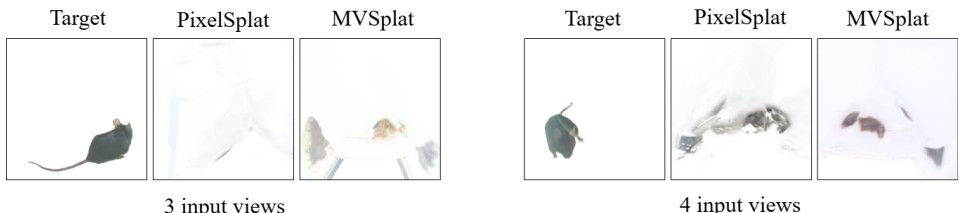

Figure 8: Results for the two feed-forward baselines with varying numbers of input views.

## G.2   TRELLIS & HunYuan3D-2

As an additional point of comparison, we tested the ability of two recent large pretrained 3D object models to render consistent and realistic mice: HunYuan3D-2 [78] and TRELLIS [65]. Both models pretrain VAEs on large collections of 3D assets and use flow matching to generate latents from conditioning images. These latents can then be decoded into meshes. Figure 9 shows ground truth conditioning views alongside the meshes produced by the both models when conditioned on these views. We note that the meshes display crisp yet often inaccurate geometric details. For example, note the third HunYuan3D-generated mesh, which has 6 legs (5 visible from rendered view) and the sixth TRELLIS-generated mesh, which has an unrealistic head shape. Furthermore, the meshes are clearly inconsistent with one another, a clear limitation of single-view conditioned models.

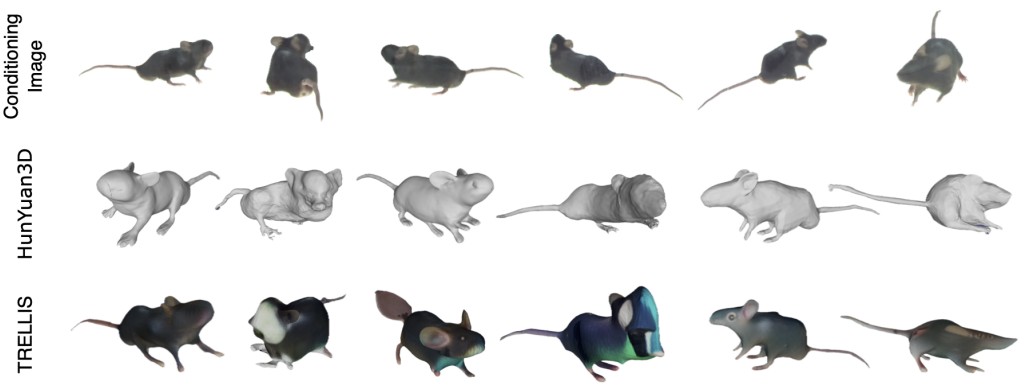

Figure 9: HunYuan3D-2 and TRELLIS produce crisp yet often inaccurate geometric detail that are inconsistent across conditioning views.

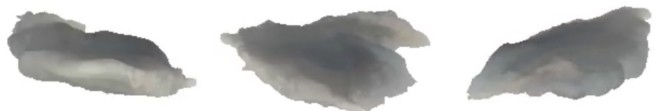

Figure 10: InstantMesh produces unrecognizable shapes given multi-view image input. Three views of the output mesh are shown. The conditioning images are shown on the first row of Figure 9.

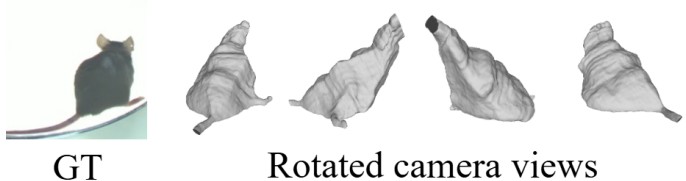

GT                    Rotated camera views

Figure 11: When the mesh generated by BANMo is rotated, it fails to reconstruct a proper shape from unseen views.

### G.3   InstantMesh

InstantMesh is a large pre-trained image to mesh pipeline that first maps a monocular image to a multi-view image using a pretrained diffusion model and then uses the multi-view images to construct a textured mesh [67]. We bypassed the first part of the pipeline and sent the multi-view images displayed in Figure 9 (top row) directly to the InstantMesh model, taking care to match the expected preprocessing steps and formats exactly. Figure 10 shows three views of the predicted mesh, which is not recognizably a mouse. We suspect that InstantMesh is unable to reconstruct a meaningful shape because of the tightly controlled range of camera parameters seen by the model during training. Specifically, the camera positions are equidistant from the object centers, which does not apply to images of a moving animal relative to stationary cameras. This design choice is reasonable for InstantMesh, which is trained on easily-manipulable synthetic 3D data, but has disadvantages for translating to messier real-world data, even in relatively well-controlled laboratory conditions.

### G.4   BANMo

We also tested BANMo [69], a well-known model for monocular 3D animal reconstruction. BANMo is a template-free approach that reconstructs animatable 3D models from monocular videos through per-video optimization. However, each sequence requires separate optimization, making the method computationally expensive and unsuitable for large-scale datasets. Moreover, its reliance on single-view input limits reconstruction accuracy, especially for occluded or rarely visible body parts. Figure 11 illustrates that BANMo struggles to reconstruct accurately under unseen views.

### G.5   VGGT & AnySplat

We tested two recent generalizable, feed-forward 3D reconstruction models, VGGT [60] and AnySplat [23], to check whether large pre-trained models could outperform or complement our lightweight model.

VGGT reconstructs point clouds and depth maps from multi-view inputs but does not produce renderings. Due to the absence of ground-truth depth or 3D meshes in our dataset, quantitative evaluation was not feasible. Qualitatively, VGGT failed to produce coherent 3D surfaces on our animal data, which features minimal overlap between views. Instead, it generated misaligned, layered point clouds, a characteristic "onion-peel" artifact also visible in elongated parts such as tails. The result is shown in Figure 12

AnySplat, which uses VGGT as its geometric backbone for Gaussian Splatting, exhibited similar issues as shown in Figure 13. While it supports direct rendering and quantitative evaluation, its zero-shot performance on our dataset was significantly lower than Pose Splatter. Fine-tuning AnySplat yielded only marginal improvements across all metrics (IoU, L1, PSNR, SSIM) and did not resolve the misalignment artifacts. The quantitative result is displayed in Table 7.

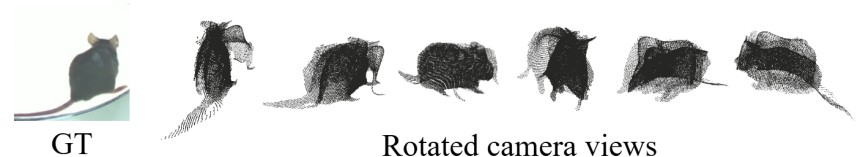

GT                                          Rotated camera views

Figure 12: Due to inaccurate camera estimation, VGGT produces a point cloud that shows an "onion-peel" layered appearance when reconstructed.

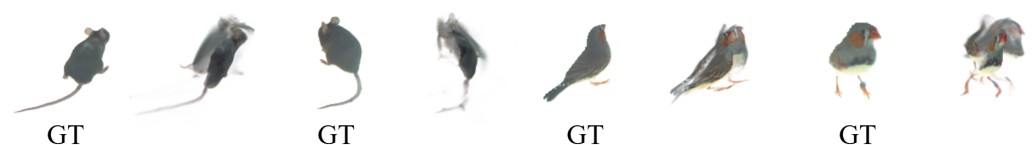

GT                      GT                      GT                      GT

Figure 13: Similar to the results of VGGT (Figure 12), AnySplat exhibits an "onion-peel" artifact and produces inaccurate reconstruction results.

These findings indicate that current generalizable models like VGGT and AnySplat struggle to handle sparse, low-overlap multi-view animal data. In contrast, a lightweight model trained from scratch, such as Pose Splatter, provides more accurate and efficient reconstructions under these challenging conditions.

## H   Additional Renderings

Figure 14 shows randomly sampled test set renderings for the three 6-camera models **from observed views**.

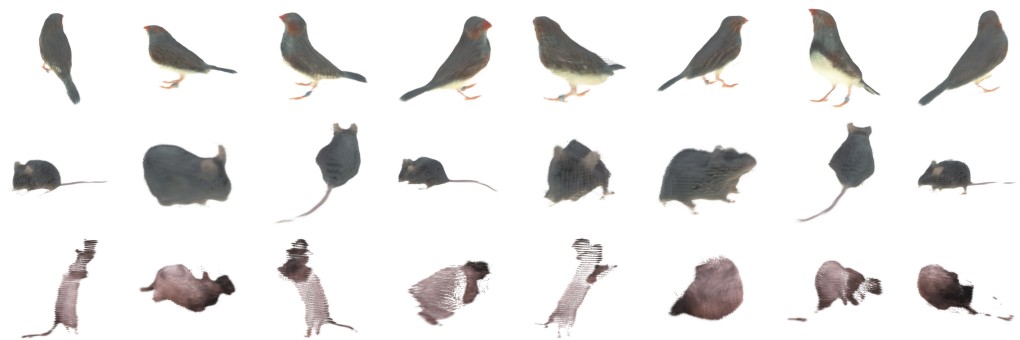

Figure 14: Representative renderings from observed views.

Figure 15 shows randomly sampled test set rendering for the three 6-camera models **from unobserved views**.

Despite the de-voxelization step in *Pose Splatter*, we observed regular grid-like patterns in many renderings. For the mouse and finch datasets we trained models with twice the spatial resolution in the voxelization step, resulting in 8 times as many voxels. As shown in Figure 16, we see a similar visual quality but no apparent regular grid patterns.

## I   Quality of 3D Keypoints

After training a SLEAP model [48] to predict 2D keypoints from images, we perform a robust triangulation across views using the known camera parameters to estimate 3D keypoints. To assess the quality of these 3D keypoints, we calculated the distribution of reprojection errors (in pixels) for

| Dataset | Views | Model | Setting | IoU↑ | L1↓ | PSNR↑ | SSIM↑ |
|---|---|---|---|---|---|---|---|
| Mouse | 5 | AnySplat | Zero-Shot | 0.403 | 1.132 | 25.1 | 0.952 |
| | | AnySplat | Fine-Tuned | 0.431 | 1.088 | 25.5 | 0.959 |
| | | Pose Splatter | – | **0.760** | **0.632** | **29.0** | **0.982** |
| | 4 | AnySplat | Zero-Shot | 0.392 | 1.227 | 25.6 | 0.961 |
| | | AnySplat | Fine-Tuned | 0.415 | 1.051 | 26.3 | 0.965 |
| | | Pose Splatter | – | **0.721** | **0.753** | **28.2** | **0.982** |
| Finch | 5 | AnySplat | Zero-Shot | 0.365 | 1.359 | 24.7 | 0.968 |
| | | AnySplat | Fine-Tuned | 0.412 | 1.174 | 25.4 | 0.971 |
| | | Pose Splatter | – | **0.848** | **0.345** | **34.5** | **0.992** |
| | 4 | AnySplat | Zero-Shot | 0.421 | 1.057 | 24.2 | 0.962 |
| | | AnySplat | Fine-Tuned | 0.459 | 1.012 | 24.9 | 0.967 |
| | | Pose Splatter | – | **0.731** | **0.685** | **29.0** | **0.981** |

Table 7: Quantitative comparison between AnySplat and Pose Splatter on Mouse and Finch datasets with 4 or 5 cameras.

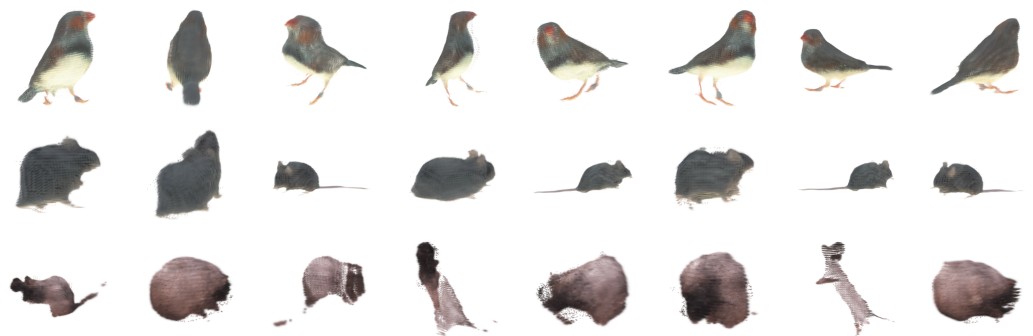

Figure 15: Representative renderings from unobserved views.

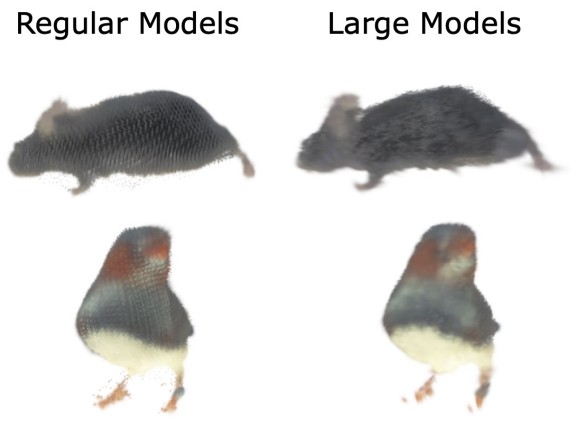

Figure 16: **Visual comparison between regular and large shape carving volumes**. The renderings on the left come are from models using a "standard" volume size of $96 \times 80 \times 64$ for the mouse and $96 \times 64 \times 80$ for the finch (the volume sizes used in the rest of the paper). The renderings on the right come from models that use a larger volume size of $192 \times 160 \times 128$ for the mouse and $192 \times 128 \times 160$ for the finch. Note that the regular grid artifacts seen on the left are not visible on the right. Best viewed zoomed in.

both the mouse and finch datasets. As seen in Figure 17, we find higher quality 3D keypoints for the

finch dataset. We believe this results from the higher quality 2D keypoints for this dataset. This may also explain the better performance predicting 3D keypoints from visual embeddings on the finch dataset (Figure 5, left). Representative examples of SLEAP-predicted 2D keypoints and reprojected 3D keypoints are shown in Figure 18.

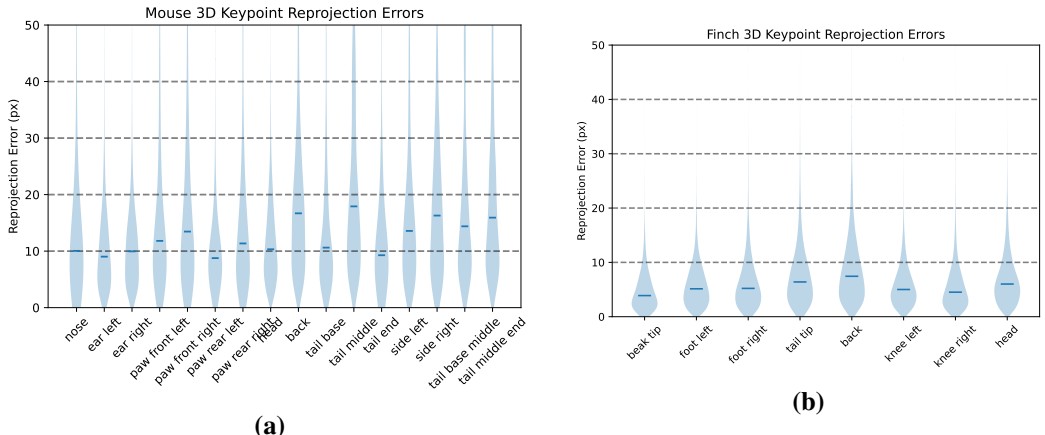

**(a)**  **(b)**

Figure 17: Summary of per-keypoint reprojection errors for **a)** mouse and **b)** finch datasets. Violin plots for each keypoint show the distribution of errors between SLEAP-predicted 2D keypoints and the 2D projections of per-frame robustly triangulated 3D keypoints. Units are Euclidean distance in pixels, for full $1536 \times 2048$ images. Median errors are marked.

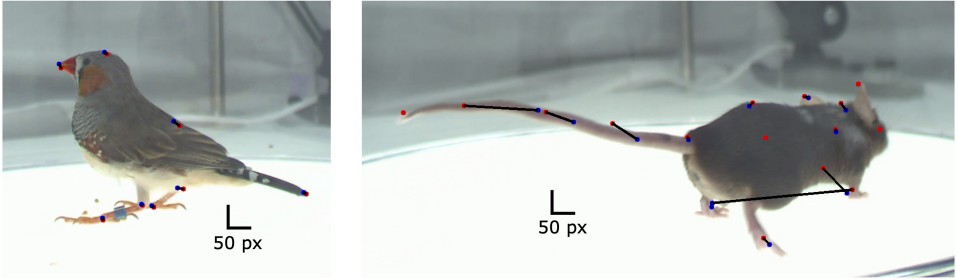

Figure 18: Frames showing SLEAP-predicted 2D keypoints (*blue*) and corresponding projected 3D keypoints (*red*). Black lines connect corresponding 2D and 3D keypoints and 50-pixel bars are shown for scale. Note the high accuracy of the triangulated finch keypoints (left) and some 2D keypoint errors for the mouse pose such as front right versus back left paw (right).

## J   Survey

To ensure transparency and enable replication of the nearest neighbor survey, with results presented in Figure 2b, we provide the survey instructions and illustrative screenshots of the user interface. The complete wording of the participant instructions is reproduced verbatim below, while Figure 19 depicts the layout of an individual survey question.

> **Please read these instructions carefully before beginning the survey.**
>
> In this study, you will judge how closely two candidate poses resemble the body posture of a reference ("query") mouse.
>
> Each question is laid out in 3 columns: the left column shows the query pose, while the center and right columns show Option 1 and Option 2. Every column contains two synchronized camera views (top and bottom) captured at the same instant, giving complementary angles on a single pose.

Your task is to decide which option—1 or 2—better matches the query. Focus only on the configuration of body parts such as the head, torso, limbs, and tail, and disregard background, lighting, or color differences. Beneath the images are two radio buttons; click the button under the option you believe is closer to the query.

There are 40 questions in total, and there is no time limit. By continuing, you confirm that you consent to your anonymous responses being used for academic research on animal-pose representations.

Question 1

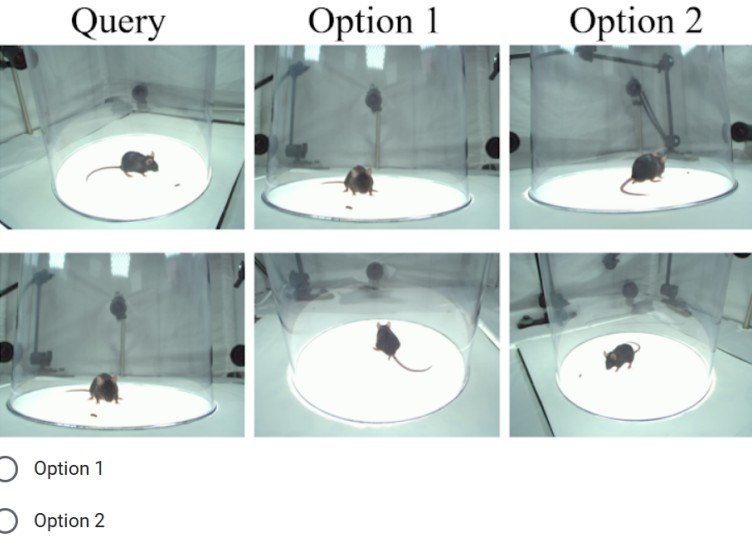

○ Option 1

○ Option 2

Figure 19: Example survey question

