# OpenReview forum: "Pose Splatter: A 3D Gaussian Splatting Model for Quantifying Animal Pose and Appearance"
_NeurIPS.cc/2025/Conference — NeurIPS 2025 poster_

### Official Review · Reviewer_r2qH · 2025-06-27

**Clarity:** 4
**Significance:** 3
**Originality:** 2
**Rating:** 5
**Confidence:** 3

**Summary:**

This paper proposes a method for estimating the 3D pose and appearance of subjects from a sparse set of views, with a focus on animal pose estimation. The approach leverages Gaussian splatting to model the 3D geometry of animals, enabling efficient feed-forward rendering of novel views. The authors introduce a low-dimensional embedding to jointly encode pose and appearance. They evaluate their method on three datasets (mice, rats, and zebra finches) demonstrating improved performance over existing Gaussian splatting techniques for 3D reconstruction and rendering, as well as over 3D landmark tracking for behavior analysis.

**Questions:**

Please address (1)-(4) above.

**Ethical Concerns:**

["NO or VERY MINOR ethics concerns only"]

**Final Justification:**

I’m positive about this work and would be happy to see it published with the revisions proposed by the authors. The authors provided comprehensive answers to most concerns raised in my and other reviews.

**Limitations:**

Fine.

**Paper Formatting Concerns:**

References should include a complete list of authors, rather than "et al.". Please revise.

**Quality:**

3

**Strengths And Weaknesses:**

I like this paper. It tackles an interesting and challenging problem and proposes a method that seems well-suited for lab setups aimed at studying animal behavior and has potential for meaningful positive impact. The paper itself is very well written and organized, and overall a pleasure to read. The pipeline is explained clearly, with both motivation and technical detail provided for each component. The work appears reproducible, and the authors provide code (which I haven’t tested) that will be made publicly available, along with two of the three datasets used in the experiments.

Below are a few concerns I would encourage the authors to address in their rebuttal, and which I believe could be resolved with a minor revision:

*(1) Overstatements:*
Some claims in the paper strike me as overstated. For example, the sentence “Pose Splatter enables analysis of large-scale, longitudinal behavior needed to map genotype, neural activity, and micro-behavior at unprecedented resolution” seems exaggerated. While improvements are demonstrated clearly, “unprecedented resolution” is misleading. The summary in lines 343–349 is more balanced and appropriate.
Similarly, claims of high-quality or high-fidelity reconstruction are made repeatedly. While the method improves upon other GS-based techniques, the visual results still appear relatively coarse due to low-resolution visual hull estimates and the inherent roughness of Gaussian splats, especially when compared to reconstructions using higher-complexity 3D representations.
Another example concerns the claimed position and azimuthal invariance. Although these properties are observed, they stem from standard PCA alignment and the phase-invariance of norm in the embedding. I recommend that the authors tone down such statements to more accurately reflect the technical contributions.

*(2) Pose Estimation:*
I have some reservations about the use of the term “pose.” In the context of animal studies, pose estimation typically involves a semantic understanding of how body parts relate to one another. The proposed method does not offer such semantic structure. Rather, it performs implicit 3D reconstruction and enables novel view synthesis. The embedding computed from these reconstructions encodes a combination of pose and appearance features that are shown to be effective for downstream tasks. However, it does not amount to pose estimation in the traditional sense. For some applications, this limitation may reduce interpretability or fail to capture the specific information needed (e.g., a small set of body landmarks might be more suitable in such cases).

*(3) Specificity to Animals:*
What exactly makes this approach specific to animals? Although the paper is framed in that context, I was left wondering whether the method is in fact general and could apply just as well to other settings, such as human subjects or other objects, where methods like 3DGS, FSGS, or GO have been successfully used. If the method is truly general, it would be useful to clarify this. If there are aspects that make it particularly well-suited for animals, those should be highlighted more explicitly.

*(4) Limitations and Practical Use:*
Although the paper mentions limitations in the conclusion, I am concerned that the method may be restricted to sterile, highly controlled environments, limiting its applicability to real lab settings. The paper does not provide enough detail on practical conditions required for the approach to work: How many views are needed? What lighting conditions are assumed? How must the cameras be arranged spatially? I encourage the authors to elaborate on these aspects to help readers understand the practical constraints of deploying the method.

---

*Additional minor issues and questions:*

* *Line 129:* Why exactly three U-Net modules? Please clarify.
* *Line 159:* Is the training loss computed over all input views, or only a subset? Please clarify.
* *Figures:* Some figure elements (especially 2(b)) are not explained until much later in the paper and feel out of context. Consider reordering or revising.
* *Line 177:* What "pretrained convolutional encoder" is used?
* *Lines 191–196:* This description is vague; more detail would help, even if included in the appendix.
*Line 219:* To better support the claimed advantage that "our model uses only about 2.5GB of GPU memory (VRAM)," it would be helpful to contextualize this number by referencing the memory requirements of competing models.
* *Figure 5 (left panel):* The content is unclear, and I couldn’t interpret what it shows. Please explain.
* *Lines 305–307:* "Example queries and answers are shown in Figure 2b, demonstrating the high degree of similarity produced by both feature sets.". The example queries and corresponding KP NN and VE NN are all quite different and I find it hard to evaluate the quality and conclusions of this experiment.
* *Lines 321–322:* The process of predicting "egocentric 3D keypoints from the visual embedding using a 5-nearest neighbor regressor" is not clear, please provide more details.

---

> ### Author Rebuttal · Authors · 2025-07-31
>
> We are grateful for your thoughtful, detailed, and positive review. Thank you for the encouraging feedback. We are especially pleased that you found the paper well-written and a pleasure to read, and that you recognized our work as a well-motivated and reproducible contribution to the study of animal behavior.
>
> Your concerns and suggestions are meaningful, and we agree that addressing them will improve the manuscript. We address each of your points below.
>
> **Overstatements:**
> Thank you for this valuable feedback. You are right that some of our claims, such as "unprecedented resolution," are overstated. Our intention was to highlight the potential for large-scale analysis enabled by our method's efficiency, but we agree the language should be more measured and precise. We have revised the last sentence of the abstract as follows:
>
> > By eliminating annotation and per-frame optimization bottlenecks, *Pose Splatter* enables analysis of large-scale, longitudinal behavior needed to map genotype, neural activity, and behavior at high resolutions.
>
> Additionally, we have revised our description of the visual embedding in the abstract:
>
> > We also propose a rotation-invariant visual embedding technique for encoding pose and appearance, designed to be a plug-in replacement for 3D keypoint data in downstream behavioral analyses.
>
>
> **Pose Estimation:**
> We agree that our method does not perform "pose estimation" in the traditional, semantic sense of identifying and structuring body parts. It is more accurately described as implicit 3D reconstruction, from which we derive a holistic descriptor of pose and appearance. We use the term "pose" in a broader sense to refer to the animal's overall physical configuration. In our revision, we will clarify this distinction, explicitly stating that our method provides an implicit or holistic representation of pose.
>
>
> **Specificity to Animals:**
> You are correct that our core pipeline is, in principle, general and could be applied to other non-rigid objects. However, it was originally designed for animal behavior because its key strengths directly solve the most significant challenges in that domain. Specifically, our method is template-free, making it immediately applicable to diverse species that lack standard 3D models like those used in human-centric research. Furthermore, it is annotation-free, bypassing the major bottleneck of defining and annotating animal keypoints. Therefore, while the algorithm is general, and could probably be used for other types of objects, its impact is probably most pronounced in the animal domain.
>
> **Limitations and Practical Use:**
> Thank you for these important practical questions. We agree that a clearer discussion of the method's settings is crucial for readers, and we will add a dedicated section on these considerations in our revised appendix. Our goal was to design a system that is robust and flexible enough for real-world lab settings. For the number of views, our method is adaptable to various multi-camera setups, performing well with a sparse set of 4 to 6 views. The approach is also robust to the kinds of non-ideal lighting found in many lab settings and does not require specialized studio lighting. This robustness is largely thanks to our reliance on a powerful foundation model (SAM2) for segmentation; because such models perform well in varied lighting, the prerequisite of a reasonable foreground mask is achievable under a wide range of typical lab conditions. Finally, the camera arrangement follows the standard best practice for any high-fidelity multi-view 3D reconstruction task: placing cameras to provide comprehensive coverage and minimize self-occlusion.
>
> **Additional minor issues and questions:**
>
> * **Three U-Net Modules (Line 129):**
> The choice of three U-Net modules was determined empirically in early experiments. We observed that performance improved incrementally as we stacked more modules, probably due to the iterative refinement of the coarse visual hull, where each module improves upon the output of the previous one. We found that three modules offered a good trade-off between performance gains, memory consumption, and forward pass time.
>
> * **Training Loss Computation (Line 159):** To clarify, the training loss is defined between the rendered output and a single, randomly selected input view for each training sample. We will add a sentence to the "Loss Terms" section to state this explicitly in the revised manuscript.
>
> * **Figure 2(b):** Thank you for this suggestion. We will revise the caption of Figure 2 and the surrounding text to provide clearer context for the nearest-neighbor study upon its first appearance, so it does not feel out of place.
>
> * **Pretrained Encoder (Line 177):** We apologize for this omission in the main text. As detailed in Appendix C, the encoder is a pre-trained ResNet-18. We will add this important detail to the main text for clarity.
>
> * **Vague Embedding Description (Lines 191-196):** The section in the main text is a summary of the final step of our embedding process, with the complete technical details of the Spherical Harmonic expansion and adversarial PCA provided in Appendix C. We have revised the main text to offer a more intuitive explanation of the process, ensuring the core concepts are easier to grasp without needing to immediately refer to the appendix:
>
> > However, we found that these feature vectors still strongly encoded the azimuthal angle of the animal, possibly due to uneven lighting conditions across views. To remove this effect, we employ an adversarial formulation of principal components analysis (PCA) to find a 50-dimensional subspace of the feature vectors that contains a large portion of the variance of the input vectors and can predict only a small portion of the variance of the sine and cosine components of the azimuthal rotation angle. We take these 50-dimensional pose descriptors as the visual embedding. See Appendix C for more details.
>
> * **Contextualizing the 2.5GB VRAM Usage (Line 219):** This is a great suggestion to better highlight our model's efficiency. As we state in the paper, our model is very lightweight, requiring only about 2.5GB of GPU memory (VRAM). For context, competing feed-forward models like PixelSplat and MVSplat consume at least **20GB** of GPU memory, depending on the dataset and batch size, while per-scene optimization models such as Gaussian Object require over **10GB**. We will add these numbers to the revised manuscript to make the practical advantages of our efficient architecture much clearer.
>
> * **Figure 5 (left panel):** This panel shows the R2 (Coefficient of Determination) score for predicting the 3D locations of each individual animal keypoint using the visual embedding. Each dot represents a keypoint (left rear paw, beak tip, etc.), and its position on the y-axis indicates the percentage of that keypoint's positional variance in an egocentric coordinate system that the visual embedding can explain using a 5-nearest neighbor regressor. See Appendix C for details. We find that the visual embedding is sufficient to explain a large portion of the positional variance of the 3D keypoints.
>
> * **Lines 305–307:** The example was chosen to illustrate a subtle difference (a head tilt to the left) that our method captured but the baseline missed. The position of the mouse in the arena is different between the Query, KP NN, and VE NN columns and so the nearest neighbor retrievals are visually distinct. However, note the leftward tilt of the head apparent in the bottom Query image. This tilt is absent in the top KP NN image, in which the mouse's head is pointed straight ahead, but visible in both VE NN images. This is the visual distinction we intended to highlight. We will improve the figure's annotations (e.g., with arrows and zoom-ins) to make the key difference more obvious.
>
> * **On the 5-Nearest Neighbor Regressor (Lines 321-322):** Thank you for pointing this out. The process is detailed in Appendix C, but we had mistakenly dropped a reference to the appendix in the part of the main text that describes this figure. To clarify: for a given test sample's visual embedding, we find the 5 training samples with the most similar embeddings (via Euclidean distance). The final predicted 3D keypoints are then simply the average of the ground-truth 3D keypoints from those 5 neighbors. We will add this explanation to the main text.
>
> * **On the Reference Formatting:** Thank you for catching this. We will expand all "et al." citations to include the complete author lists in the final version, adhering to the conference's formatting guidelines.

---

> > ### Comment · Reviewer_r2qH · 2025-08-05
> >
> > Thank you for the detailed responses. I’m positive about this work and would be happy to see it published with the revisions proposed by the authors.

---

### Official Review · Reviewer_mWtm · 2025-07-01

**Clarity:** 3
**Significance:** 2
**Originality:** 2
**Rating:** 4
**Confidence:** 4

**Summary:**

This paper proposes Pose Splatter, a method that leverages shape carving and Gaussian splatting to reconstruct laboratory animals, such as mice. The authors extract rotation-invariant visual embeddings based on Pose Splatter for animal behavior analysis. Experiments demonstrate the reconstruction capability and validate that the extracted visual embeddings can be used to understand animal behaviors.

**Questions:**

Please see weaknesses.

**Ethical Concerns:**

["NO or VERY MINOR ethics concerns only"]

**Final Justification:**

In the first stage, my major concerns are:

1. Comparison with object-level reconstruction methods (InstantMesh, TRELLIS).

2. Limited improvements compared with GaussianObject.

3. Comparison with action recognition models.

4. Comparison with feed-forward methods based on VGGT.

5. Limited training and testing data, restricted to mice, pigeon, and finches.

After the rebuttal, concerns 1–4 have been mostly addressed. My remaining concern is the limited data.
The authors say they will release this dataset as part of their contribution, which I consider valuable given the difficulty of obtaining such data.
As a result, I am inclined to raise my score to 4.

**Limitations:**

Yes

**Paper Formatting Concerns:**

No formatting issues

**Quality:**

3

**Strengths And Weaknesses:**

Strengths
1. The paper is well-structured and easy to follow.
2. The idea of extracting visual embeddings based on a reconstruction model for animal behavior analysis is reasonable and well-motivated. The pipeline is clearly constructed and logically justified.

Weaknesses
1. The authors primarily compare their method with scene-level reconstruction approaches, such as MVSplat and PixelSplat. However, since the focus is on modeling individual animals, it would be more appropriate to include comparisons with object-level feed-forward models, such as GRM [1], TRELLIS [2], and HunYuan3D [3]. These models do not require species-specific templates and are known for their strong generalization capabilities.

2. The dataset used in this study is quite limited, incorporating only two types of animals: mice and finches. This raises concerns about the generalizability and overall effectiveness of the proposed method. Moreover, the observed improvement over the per-scene optimization method (GO) appears marginal, which further questions the necessity and impact of employing a feed-forward model in this setting.
Existing feed-forward models are typically designed for large-scale datasets and are valued for their strong generalization capabilities across diverse samples. Given the limited data available in this study, it is worth carefully reconsidering whether training a feed-forward model from scratch is truly justified.
In fact, leveraging the priors inherent in well-established base models may yield better performance and efficiency than training a new model from scratch on such a small dataset. Additionally, expanding the dataset to include a broader range of animal species would substantially strengthen the validation of the proposed approach and better demonstrate its potential for wider applicability.
3. For understanding animal behavior, action recognition models that process video or image sequences are also widely used. The paper does not clearly explain the advantages of the proposed method over these established action recognition approaches. Including additional comparative experiments or analyses would be helpful to clarify the unique contributions and practical benefits of the proposed method in behavioral analysis.

[1] GRM: Large Gaussian Reconstruction Model for Efficient 3D Reconstruction and Generation

[2] TRELLIS: Structured 3D Latents for Scalable and Versatile 3D Generation

[3] Hunyuan3D: https://github.com/Tencent-Hunyuan/Hunyuan3D-2

---

> ### Author Rebuttal · Authors · 2025-07-31
>
> We sincerely thank the reviewer for the positive feedback on our paper's structure, motivation, and logical pipeline. We are glad you found the idea of using visual embeddings for behavior analysis to be reasonable and well-motivated.
>
> We would like to address the weaknesses you raised, as we believe they may stem from a misunderstanding of our work's primary task, goals, and contributions.
>
> **Comparisons to Object Models (GRM, TRELLIS, HunYuan3D):**
> Thank you for suggesting these powerful recent models. Unfortunately, the code for GRM has not been publicly released, making a direct comparison impossible. Regarding TRELLIS and HunYuan3D, while they are indeed powerful, they are designed for a fundamentally different task. These are generative models trained on massive datasets of 3D objects to create novel 3D objects from inputs like **text or a single image**. In contrast, our method is a reconstruction model designed for scientific applications that require the model to _accurately_ recover the specific pose and appearance of an animal from a sparse set of views, _where no ground truth 3D structure is available_.
>
> Despite this task mismatch, we took your suggestion and tested these models on our data. The results highlighted a critical issue: these models tend to hallucinate occluded parts rather than reconstruct them. For instance, they might invent a symmetrical backside for an animal when it's not visible in the input view. This leads to the generation of a plausible 3D animal, but not an accurate reconstruction of that animal's true size, shape, and posture in that specific moment. Because our goal is the precise 3D reconstruction needed for scientific behavioral analysis, this tendency to invent data makes these otherwise powerful generative models unsuitable for our specific application.
>
> **Comparison with GaussianObject:**
> Regarding the comparison with GaussianObject (GO), while our method does not offer a dramatic improvement in final rendering quality, the motivation for our feed-forward approach was never about marginal gains in image metrics; it was about enabling scalability for large-scale scientific analysis. As we report in the paper, GO requires roughly one hour of optimization per scene, while our method's inference time is just 30 milliseconds—a speedup of over 100,000 times. For the behavioral neuroscience domain, where datasets often contain hundreds of thousands of frames, this difference is critical. Per-scene optimization is computationally prohibitive at this scale, making a fast, feed-forward model the only viable path for practical, large-scale analysis.
>
> **Training from Scratch vs. using a Pre-Trained Model:**
> We appreciate the reviewer’s suggestion to use a pre-trained model, but a suitable “well-established base model” for our specific task does not yet exist. The models mentioned, such as TRELLIS and HunYuan3D, are designed for generating objects from single views or text and are not optimized for reconstruction from the kind of sparse, low-overlap multi-view data we use. In the absence of a pre-trained foundation model for our task, training from scratch is the only viable option.
>
> Fortunately, our model is computationally lightweight, requiring only about 2.5GB of VRAM and a few hours of training time per dataset, making this approach highly accessible to researchers. Despite this efficiency, our model demonstrates strong cross-species generalization, as shown in Figure 3a and our quantitative results in Section 4.3. We therefore believe our approach is a practical and effective solution until large-scale foundation models for this specific task are developed.
>
> Regarding the request to add more species beyond the three in the current work, we agree this would strengthen the paper and it is a goal for future work. As suggested by reviewer peFs, we are currently working to apply _Pose Splatter_ to additional multiview videos with pigeons and monkeys. However, we wish to emphasize that publicly-available synchronized, multi-view data of freely behaving animals is relatively rare. We believe our current contribution of two new datasets that will be released publicly, is already a significant and impactful contribution to the field.
>
> **Video vs. Image Models**
> This is an insightful question that gets to the heart of our contribution. Traditional action recognition models are fundamentally temporal: they process a sequence of frames to classify an action that unfolds over time. In contrast, our method is static: its primary function is to create a high-fidelity 3D representation of an animal's posture at a single, specific timestep. The advantage of our approach is the superior quality of this per-timestep representation, which serves as a better building block for any downstream temporal analysis. Additionally, the static model gives us the flexibility to perform filtering or smoothing as required by specific scientific questions.
>
> Therefore, the unique contribution of our work is not to replace action recognition models, but to provide them with a richer, more informative input. By accurately describing **how** the animal is posed at every moment, we enable more powerful downstream models to determine **what** action is occurring over time.

---

> > ### Comment · Reviewer_mWtm · 2025-08-03
> >
> > Thank you for the detailed feedback. The concerns raised under Weakness 3 have been addressed.
> >
> > 1. Apologies for the unreleased code (GRM). As an alternative, would InstantMesh[1] be a viable option?
> >
> > 2. Regarding Weaknesses 1 and 2, would it be feasible to explore VGGT-based or DUST3R-based methods in this domain? These approaches may offer promising directions for improving performance and robustness.
> >
> >
> > [1] InstantMesh: Efficient 3D Mesh Generation from a Single Image with Sparse-view Large Reconstruction Models

---

> > > ### Author Response · Authors · 2025-08-06
> > >
> > > **Regarding InstantMesh:**
> > >
> > > Thank you for suggesting InstantMesh; it is indeed a very impressive and efficient model. Its core mechanism, which uses a diffusion model to **generate** a set of plausible multi-view images from a single input and then builds the 3D mesh from those generated views, is fundamentally different from our task. Our work is focused on multi-view **reconstruction**, where the objective is to recover the precise 3D shape and posture of a specific animal at a specific moment in time using a sparse set of actually captured, synchronized images.
> > >
> > > Following your advice, we tested InstantMesh on our data, and the results confirmed our concerns. Similar to HunYuan3D and TRELLIS, the model's tendency to "hallucinate" or invent plausible but geometrically inaccurate details for occluded parts makes it unsuitable for our scientific application. This is because our downstream analysis requires a high-fidelity reconstruction of the animal's true posture, and these invented details introduce unacceptable inaccuracies.
> > >
> > > Therefore, while InstantMesh is state-of-the-art for its intended generative task, it is not an appropriate baseline for the scientific reconstruction problem we are addressing.

---

> > > > ### Author Response · Authors · 2025-08-06
> > > >
> > > > **Generalizable feed forward models:**
> > > >
> > > > We thank the reviewer for the insightful suggestion to explore more powerful generalizable models. Following this advice, we conducted new experiments with VGGT, which is currently a state-of-the-art generalizable, feed-forward, multi-view 3D reconstruction tool, as well as AnySplat (Jiang, L., Mao, Y., Xu, L., Lu, T., Ren, K., Jin, Y., ... & Dai, B. (2025). AnySplat: Feed-forward 3D Gaussian Splatting from Unconstrained Views. arXiv preprint arXiv:2505.23716)  [1], a recent Gaussian Splatting model built upon it.
> > > >
> > > > First, we tested VGGT on our data. Since VGGT reconstructs point clouds and depth maps, but not renderings, and our dataset lacks 3D ground truth or ground-truth depth, a direct metric-based evaluation was not possible. However, our qualitative analysis revealed a significant challenge. Despite being a generalizable model, VGGT struggled to produce accurate point clouds, perhaps because our real-world animal data, which features minimal overlap between views, is substantially more difficult than typical scene or object datasets. We observed that the point clouds from individual views were incorrectly superimposed rather than forming a single, coherent surface, resulting in an "onion-peel" effect of misaligned, layered surfaces. This effect was also observed in the mouse tail, resulting in several overlapping tails corresponding to the different views.
> > > >
> > > > This tendency naturally persisted in AnySplat, which uses a pretrained VGGT as its backbone to enable generalizable, feed-forward Gaussian Splatting. Unlike VGGT, AnySplat can render 2D images from 3D objects directly via Gaussian Splatting. This allowed us to perform a direct, quantitative comparison using the same PSNR, IoU, L1, and SSIM metrics we used to evaluate Pose Splatter. The results of our comparison with AnySplat are presented in the table below with Pose Splatter's results from Table 2a added for comparison.
> > > >
> > > > **Mouse**
> > > > |       | IoU↑ | L1↓ | PSNR↑ | SSIM↑ |
> > > > |:-----:|:----:|:---:|:-----:|:-----:|
> > > > | AnySplat (5 cam) |  0.403 | 1.132 |  25.1  |  0.952  |
> > > > | PoseSplatter (5 cam) |  **0.760** | **0.632** |  **29.0**  |  **0.982**  |
> > > > | | | | | |
> > > > | AnySplat (4 cam)|  0.392 | 1.227 |  25.6  |  0.961  |
> > > > | PoseSplatter (4 cam) |  **0.721** | **0.753** |  **28.2**  |  **0.982**  |
> > > >
> > > > **Finch**
> > > > |       | IoU↑ | L1↓ | PSNR↑ | SSIM↑ |
> > > > |:-----:|:----:|:---:|:-----:|:-----:|
> > > > | AnySplat (5 cam) |  0.365 | 1.359 |  24.7  |  0.968  |
> > > > | PoseSplatter (5 cam) |  **0.848** | **0.345** |  **34.5**  |  **0.992**  |
> > > > | | | | | |
> > > > | AnySplat (4 cam) |  0.421 | 1.057 |  24.2  |  0.962  |
> > > > | PoseSplatter (4 cam) |  **0.731** | **0.685** |  **29.0**  |  **0.981**  |
> > > >
> > > > As the results show, *Pose Splatter* consistently outperforms AnySplat by a large margin across all metrics. We observe the 3D Gaussian surfaces predicted from different views failed to fuse into a single object; instead, they remained as distinct, misaligned layers, just like VGGT's point clouds. Critically, providing additional views did not mitigate this issue. It often exacerbated it by adding more misaligned surfaces to the scene, leading to no improvement in IoU or PSNR scores. These findings suggest that simply applying a large, generalizable model is not yet an effective strategy for high-fidelity 3D animal reconstruction from sparse views. Our experiments indicate that training a tailored model from scratch is a more robust approach.
> > > >
> > > > With that said, we agree with the reviewer's insight that VGGT still holds potential for pose-free, feed-forward 3D animal reconstruction. Its ability to produce relatively accurate 3D attributes from uncalibrated images in a single pass is a powerful starting point. While our experiments show it is not sufficiently accurate when used "out-of-the-box" on our data, a promising future direction would be to use VGGT as an initial estimator and then explicitly build a module to refine its predicted 3D attributes and camera poses. We believe this approach could pave the way for a highly accurate, yet still pose-free, feed-forward system. In a similar vein, an interesting area of future work would be to leverage the learned 3D shape priors from 3D object generation models such as TRELLIS, HunYuan3D, or InstantMesh in concert with multi-view data to enable more accurate 3D reconstructions.
> > > >
> > > > [1] AnySplat: Feed-forward 3D Gaussian Splatting from Unconstrained Views

---

> > > > > ### Comment · Reviewer_mWtm · 2025-08-07
> > > > >
> > > > > Thank you for the detailed feedback.
> > > > >
> > > > > 1. You can directly input your multi-view images into InstantMesh, bypassing the single-view to multi-view stage.
> > > > >
> > > > > 2. Do you finetune AnySplat on your training data? If not, the comparison may not be entirely fair.

---

> > > > > > ### Author Response · Authors · 2025-08-08
> > > > > >
> > > > > > Thank you for your valuable comments.
> > > > > >
> > > > > > **Multi-view InstantMesh**
> > > > > >
> > > > > > This is a great suggestion. To test the multi-view capabilities of InstantMesh, we bypassed the multi-view diffusion step of InstantMesh and passed 6 cropped views and corresponding camera parameters corresponding to each cropped view to the model, taking care follow the data preparation of the multi-view diffusion pipeline. Qualitatively, we found the inferred meshes have generally correct colors, but are unrecognizable as animals, with indistinct blob-like shapes.
> > > > > >
> > > > > > We suspect that this multi-view InstantMesh is unable to reconstruct a meaningful shape because of the tightly controlled range of camera parameters seen by the model during training. Specifically, the camera positions are equidistant from the object centers, which does not apply to images of a moving animal relative to stationary cameras. This design choice is reasonable for InstantMesh, which is trained on easily-manipulable synthetic 3D data, but has disadvantages for translating to messier real-world data, even in relatively well-controlled laboratory conditions.

---

> ### Author Response · Authors · 2025-08-08
>
> **Finetuned AnySplat**
>
> The initial results we presented were from testing the pre-trained AnySplat model directly, without any fine-tuning. We took this approach to directly answer your original question: whether leveraging the prior of a large, generalizable model would be more performant or efficient than training a lightweight, feed-forward model like Pose Splatter from scratch. From an efficiency standpoint, we believe a true "apples-to-apples" comparison necessitates using the pre-trained model as-is. The moment a large model requires fine-tuning, a process that for AnySplat demands over 40GB of VRAM and significant training time, it loses its primary advantage of efficiency over a model like Pose Splatter, which trains from scratch in a few hours with less than 3GB of VRAM. Additionally, the primary experimental results presented in the AnySplat paper are zero-shot evaluations. Therefore, our initial zero-shot experiment was designed to test the practical, out-of-the-box utility of these large models.
>
> Nonetheless, we agree that it is a valuable experiment to see how much performance could be gained by fine-tuning AnySplat on our data. Following your suggestion, we proceeded with this experiment. We adhered to the recommended training protocols from the official AnySplat paper and codebase, stopping the fine tuning process once the validation loss plateaued and key validation metrics showed no further improvement.
>
> The results are as follows:
>
> **Mouse**
> |       | IoU↑ | L1↓ | PSNR↑ | SSIM↑ |
> |:-----:|:----:|:---:|:-----:|:-----:|
> | AnySplat (Zero-Shot, 5 cam) |  0.403 | 1.132 |  25.1  |  0.952  |
> | AnySplat (Fine-Tuned, 5 cam) |  0.431 | 1.088 |  25.5  |  0.959  |
> | PoseSplatter (5 cam) |  **0.760** | **0.632** |  **29.0**  |  **0.982**  |
> | | | | | |
> | AnySplat (Zero-Shot, 4 cam)|  0.392 | 1.227 |  25.6  |  0.961  |
> | AnySplat (Fine-Tuned, 4 cam)|  0.415 | 1.051 |  26.3  |  0.965  |
> | PoseSplatter (4 cam) |  **0.721** | **0.753** |  **28.2**  |  **0.982**  |
>
> **Finch**
> |       | IoU↑ | L1↓ | PSNR↑ | SSIM↑ |
> |:-----:|:----:|:---:|:-----:|:-----:|
> | AnySplat (Zero-Shot, 5 cam) |  0.365 | 1.359 |  24.7  |  0.968  |
> | AnySplat (Fine-Tuned, 5 cam) |  0.412 | 1.174 |  25.4  |  0.971  |
> | PoseSplatter (5 cam) |  **0.848** | **0.345** |  **34.5**  |  **0.992**  |
> | | | | | |
> | AnySplat (Zero-Shot, 4 cam) |  0.421 | 1.057 |  24.2  |  0.962  |
> | AnySplat (Fine-Tuned, 4 cam) |  0.459 | 1.012 |  24.9  |  0.967  |
> | PoseSplatter (4 cam) |  **0.731** | **0.685** |  **29.0**  |  **0.981**  |
>
> The results clearly show that while fine-tuning provides a marginal improvement, it does not dramatically boost AnySplat's performance on our dataset. We believe this is because AnySplat's architecture is fundamentally dependent on the geometric priors distilled from its pre-trained VGGT backbone. An examination of its loss function confirms a heavy reliance on these priors (e.g., for depth and camera parameters). As noted in the AnySplat paper (Table 4), the improvement in camera pose estimation over VGGT, a critical component for accurate 3D reconstruction, is limited.
>
> Qualitative analysis of our fine-tuned results confirms this dependency. The characteristic artifacts of VGGT, such as the "onion-peel" effect where 3D surfaces from different views are inaccurately layered due to imprecise camera poses, remain largely unresolved after fine-tuning.
>
> Therefore, we conclude that for challenging datasets like ours, which feature minimal overlap between views, simply applying or even fine-tuning pre-trained large models offers limited returns. These models remain heavily constrained by the priors of their backbones, preventing dramatic performance gains on out-of-distribution data. In such cases, training a lightweight, feed-forward model like Pose Splatter from scratch proves to be a more effective strategy for achieving high-fidelity results.
>
> Furthermore, the significant computational cost of fine-tuning AnySplat (requiring >40GB of GPU memory and over 24 hours of training) compared to Pose Splatter (<3GB memory, a few hours of training) reinforces our claim that Pose Splatter is more computationally efficient for this task.

---

> > ### Comment · Reviewer_mWtm · 2025-08-09
> >
> > Thank you for the detailed feedback.

---

### Official Review · Reviewer_P9fo · 2025-07-02

**Clarity:** 2
**Significance:** 3
**Originality:** 3
**Rating:** 4
**Confidence:** 4

**Summary:**

This paper proposes Pose Splatter, which leverages shape carving and 3D Gaussian splatting to model animal pose and appearance.  Specifically, the foreground masks for multi-view images are back-projected to generate a rough shape estimation, followed by a stacked 3D U-Net to refine the coarse volume and a MLP to predict Gaussian parameters.  The whole network is trained with image based and mask based losses. Moreover, a visual embedding technique is also introduced to provide descriptors of pose and appearance for downstream analysis tasks.

**Questions:**

Please refer to the weaknesses.

**Ethical Concerns:**

["NO or VERY MINOR ethics concerns only"]

**Final Justification:**

I raise my score to borderline accept after discussing with other reviewers. The authors are encouraged to claim more on the contribution to the animal behavior domain instead of technical contribution if the paper gets accepted.

**Limitations:**

Yes

**Quality:**

2

**Strengths And Weaknesses:**

Strengths
1. The paper tackles an interesting direction of leveraging 3D GS to model animal pose and shape information for downstream analysis.
2. The proposed method works well on the mouse and finch dataset, and achieves cross-category generalization

Weaknesses
1. The technical contribution of the proposed method is limited. The pipeline consists of shape-carving based coarse estimation and 3D U-Net based refinement, these are quite standard and lack novelty
2. For quantitative comparison, the proposed Pose splatter only compares with PixelSplat and MVSplat, which are designed for general scene. More relevant baselines such as generalizable human gaussian [A, B] should be included in the comparison. Moreover, both PixelSplat and MVSplat uses two views, which is a bit unfair.

A. Junjin Xiao et al. RoGSplat: Learning Robust Generalizable Human Gaussian Splatting from Sparse Multi-View Images. CVPR 2025

B. Youngjoong Kwon et al. Generalizable Human Gaussians for Sparse View Synthesis. ECCV 2024.

3. The description of both the visual embedding technique and the corresponding experimental design is not very clear.

---

> ### Author Rebuttal · Authors · 2025-07-31
>
> We thank the reviewer for the feedback. We are glad you found the direction of our work interesting and noted our strong performance on the animal datasets, including cross-category generalization.
>
> We understand your concerns regarding technical contribution, experimental comparisons, and clarity. We believe these points may stem from a misunderstanding of our paper's core objective and the specifics of our evaluation. We would like to clarify these points below.
>
> **Standard Components:**
> We respectfully disagree with the assessment that our work has limited technical novelty. While the individual components like shape carving and U-Nets are established, the core innovation of our work lies in their integration into a new, end-to-end framework that solves a problem existing methods do not: the annotation-free 3D reconstruction of diverse animals. Our primary contribution is not a new network layer, but a complete system that works "without prior knowledge of animal geometry, per-frame optimization, or manual annotations." This is a fundamental departure from methods that require species-specific templates or costly per-scene optimization. Additionally, the standard components in our work enable a fast, lightweight, high-performing solution that requires only 2.5 GB VRAM, compared to about 20 GB VRAM for PixelSplat and MVSplat. Furthermore, we introduce a novel pose quantification tool—the visual embedding—which we demonstrate outperforms traditional 3D keypoints in both human preference studies and downstream behavior prediction tasks. The other reviewers (peFs, r2qH) recognized the novelty and impact of both our overall system and the visual embedding, affirming that it is a significant contribution to the field of animal behavior analysis.
>
>
> **Choice of Comparison Baselines:**
> We would like to clarify our choice of baseline comparisons, as this seems to be a point of misunderstanding. The suggested baselines, such as RoGSplat and Generalizable Human Gaussians, are explicitly designed for human reconstruction and rely heavily on human-specific priors like the SMPL body model, which are unavailable for our animal subjects. In fact, to even attempt a comparison, one must first check if it's possible to fit SMPL parameters to the animal data. We tried this, but the SMPL model failed to converge on our animal datasets, optimizing into distorted shapes that did not reflect the animals' true forms. Also, the central premise of Pose Splatter is its ability to work without any species-specific templates; therefore, comparing our general animal model to a human template-based model would be an inappropriate comparison. We chose PixelSplat and MVSplat precisely because they are generalizable, feed-forward models not tied to a specific category. While these models are generally used for reconstructing whole scenes, they are also known for reconstructing individual objects in the scene well and represent the current state-of-the-art for feed-forward, generalizable 3D Gaussian splatting.
>
> Also, regarding the use of two views for these baselines, this was not an arbitrary or unfair limitation. We were following the original authors' recommendations, as stated in our experiment result section (Section 4.3.): "In line with each author's recommendations, we trained both PixelSplat and MVSplat with two input views." Furthermore, our experiments in Appendix G show that providing these models with more views did not improve their performance. These models are designed for high-overlap stereo pairs and struggle in sparse-view settings with minimal overlap like ours, regardless of the number of input views, as stated in Section 4.3.
>
> We respectfully ask the reviewer to consider that our paper's primary goal is not to achieve a new state-of-the-art performance in general 3D reconstruction, but to propose a novel and practical system for animal pose quantification. The model comparison serves as a reference point, but our main contribution is twofold: first, achieving annotation-free 3D reconstruction (a task not addressed by previous animal pose models) and second, introducing the visual embedding as a new tool for behavioral analysis that outperforms sparse keypoint methods.
>
> **Unclear Description of Visual Embedding:**
> As other reviewers did not raise this concern, we would be happy if you could point to any specific areas of confusion so we can address them directly in our revision. In the meantime, we hope a concise summary below helps clarify our method, which is also detailed across the main text and appendices.
>
> The process involves rendering images of the 3D model from 32 virtual cameras on a sphere, passing them through a pre-trained ResNet-18 encoder to get 512-dimensional features, applying a Spherical Harmonic expansion and taking the squared magnitude of the coefficients to achieve rotation-invariance, and finally using adversarial PCA to produce the final 50-dimensional embedding. The experiments validating this embedding, including the human preference study and the behavior prediction task, are detailed in Section 4.3, with additional specifics in Appendices C and H.

---

> > ### Comment · Reviewer_P9fo · 2025-08-06
> >
> > Thank the authors for the responses. Regarding the two key contribution the authors mentioned: achieving annotation-free 3D reconstruction (a task not addressed by previous animal pose models) and second introducing the visual embedding as a new tool for behavioral analysis that outperforms sparse keypoint methods.  Firstly, there are works working on annotation-free 3D animal reconstruction such as BANMo, which is also template-free and only requires casual videos as input. Secondly, the proposed visual embedding is interesting, however, the two contributions are quite separated. The visual embedding can be obtained as long as the 3D model is available, not necessary depend on the proposed 3D reconstruction pipeline.

---

> > > ### Author Response · Authors · 2025-08-06
> > >
> > > **On Annotation-Free Animal Reconstruction and BANMo:**
> > >
> > > We appreciate the reviewer for bringing up BANMo. This method is indeed annotation- and template-free. However, we wish to respectfully highlight two important distinctions in both methodology and application that define our contribution's novelty.
> > >
> > > First, BANMo operates via **per-video optimization**, meaning it is not a feed-forward model. The process of learning a detailed, canonical model from a single monocular video sequence is computationally expensive, taking several hours to optimize for even a short clip of a few dozen seconds. A model optimized on one video cannot be generalized to another; it must be re-optimized from scratch for each new sequence. Consequently, such optimization-based methods are impractical for the large-scale scientific studies our work is designed to support, which often involve massive datasets of video recordings.
> > >
> > > Second, BANMo's reliance on **monocular video** fundamentally limits its reconstruction accuracy. While it cleverly aggregates 3D shape information over time, making it more robust than typical single-image reconstruction, its accuracy is still not as reliable as methods that use multi-view data. Our tests with BANMo on our dataset confirmed this limitation. Body parts that were not clearly or consistently visible in the monocular video were reconstructed with significant geometric inaccuracies and artifacts.
> > >
> > > Therefore, while BANMo is indeed annotation-free, it is computationally inefficient due to its per-video optimization and geometrically limited by its single-view nature. We believe our primary contribution, as stated in introduction, a model that is simultaneously **annotation-free, template-free, and feed-forward,** remains valid and significant, addressing a clear gap not filled by optimization-based or monocular methods like BANMo.
> > >
> > > **On the Relationship Between Our Reconstruction Pipeline and the Visual Embedding:**
> > >
> > > The reviewer is correct that the visual embedding technique could, in principle, be applied to any 3D model that admits rendering from arbitrary viewpoints, and does not rely on the specifics of the proposed model. However, allow us to detail the reasons we believe the visual embedding and the 3D model perform complementary roles and form a coherent contribution.
> > >
> > > Consider our primary motivation of supporting the scientific study of laboratory animals. 3D keypoint descriptors are the current state-of-the-art method for precisely quantifying animal pose in this setting, and their moderate dimensionality ($d \approx 50$) enable convenient and flexible downstream analyses (for example, quantifying the effects of a drug on behavior). The visual embedding technique was designed to produce moderate dimensional pose and appearance descriptors that, while providing more complete descriptions of pose and appearance, allow the same convenience of 3D keypoints and can replace them in existing workflows.
> > >
> > > While the visual embedding is the final output that enables downstream scientific analyses, our reconstruction pipeline is the engine that makes it possible. Before our work, there was no good option to generate the required 3D animal models with the speed, precision, and generality needed for large-scale studies. One could not simply obtain a 3D model to create these embeddings; existing methods were either too slow (per-scene/per-video optimization) or required non-existent species-specific templates. The proposed model was expressly designed to overcome this bottleneck.
> > >
> > > Therefore, our contribution is not just the embedding method in isolation, but the entire system that makes its application feasible and scalable. We hope this clarifies how our contributions are both distinct from prior works like BANMo and form a single, practical system. We appreciate the reviewer's critical feedback, which has helped us sharpen our manuscript's core message.

---

> > > > ### Comment · Reviewer_P9fo · 2025-08-07
> > > >
> > > > Thank the authors for clarification.  I will further discuss with other reviewers about the technical contribution of the paper before I make the final decision.

---

### Official Review · Reviewer_peFs · 2025-07-02

**Clarity:** 2
**Significance:** 2
**Originality:** 3
**Rating:** 4
**Confidence:** 4

**Summary:**

The article presents a novel approach, pose splatter, for computing 3D pose and appearance using shape carving and 3D Gaussian splatting. The method is novel in terms of predicting 3D pose without knowledge of animal body geometry or manual annotation. The paper also introduces a novel technique for encoding pose, which is designed to replace keypoint based pose information. This step will help in behaviour analysis based on pose of the animal. The method performs better than state of the art in terms of capturing all the subtle details of the pose, at the same time the pose computation time is reduced because per-frame optimization is not needed, similarly the method is helpful for large scale deployment because annotation of keypoint is not necessary.

**Questions:**

- The authors should explain limitations very clearly (refer weakness above and limitations section below)
- There are 3 multi-view datasets (not cited) with ground truth on 3D posture, (ref wekness) why they are not suitable for this approach?
- The metrics have to be clearly explained in the paper, in a way that a biologist would be able to use the method, if the method is intended for use of biologists. Anyhow, current effort makes it unclear for any reader even in the AI community.
- If there are camera limitations or volume limitations, what type of experiments would be suitable for this method or currently the method is an improvement over existing method but not ready to be deployed or tested in context of behavior experiments.
- While pose estimation is requirement for modern experiments, all datasets used are with 6 cameras, please explain if this is common setup for neurological experiments? Currently, the setup looks customized for the experiment. This is not a limitation but clarification would be helpful for readers with non-biology background.
- The two datasets seem to be introduced in the paper, or they are publicly available? If not, do authors intend to release them? If yes, the datasheet for dataset (a common practice at NeurIPS datasets) should be compiled and added, in order to have clarity over how the data is collected and what is it’s relevance to proposed downstream applications.

**Ethical Concerns:**

["NO or VERY MINOR ethics concerns only"]

**Final Justification:**

The authors did sufficient work to resolve my queries. The editing of text is something we have to take their word on it but they have demonstrated that their work does scale to other experimental scenarios relevant for fundamental science within and outside domain of neurology. Involving such methods is great for NeurIPs community as such.

Addition of the pigeon dataset made the paper much broader. Event if results are not great, camera based setups are more commonly being used commonly for behaviour studies. In this context, 3D pose is becoming much more crucial for high resolution behaviour analysis. From this perspective, I believe the paper must be given support.

I have not updated the score because the impact is strong on application domain than technical domain but I still maintain that reasons to accept are more than rejecting, the field of application is extremely important and difficult subject of animal pose does deserve more attention.

**Limitations:**

The authors do not have a clear limitation section, this makes it difficult to understand the limits of the given method e.g. how many cameras are best to use, does the method work with more than one animal (it is fine if it does not but still good to know what authors think), does it work with sequences in the wild?, Why existing multiview datasets are not sufficient?

The authors wrote in the checklist that camera based limitations are mentioned but there is not clear statement about it in the paper. Please clarify these details since they are important for practical consideration.

Also, the claim that keypoint based methods require intensive labor is true but if the existing method required X no of cameras, where as classical stereo methods can do the job with 2 cameras the users have a choice to make, especially in wild settings where multiple calibrated cameras is a problem. This is also a problem where animals have to be studies in a large volume, if the dependecies of method is in a certain volume it has to be mentioned. Currently, all of these details remain a bit unclear.

**Quality:**

3

**Strengths And Weaknesses:**

Strength :

- The paper picks up on an interesting problem and it is challenging, animal posture computation is highly sought after tool for automating behavior experiments. Posture is important information for extracting behaviors or activities.
- The proposed method brings as new dimension to 3D pose estimation of animals, traditionally used with either keypoint triangulation or mesh based approached. This approach goes away from it, which simply makes it easier for researchers to deploy the solutions without providing initial information or worrying about species specific geometry.
- The authors have tested the method with multiple datasets, and show the power of method in terms of accuracy in prediction of behavior and show that it is an improvement over keypoint based methods in most cases for behavior prediction.
- The authors have captured related work for the most part and explained the method in detail.

Weakness:

- The introduction is a bit unclear about following aspects in terms of being specific, the number of cameras required for the approach is not mentioned ( “A small set of calibrated cameras”).
- While related work in the field is mentioned extensively, including the outdoor / wild applications, there are three datasets which are fitting for the purpose but not mentioned or covered 3D-POP (pigeons in indoor), 3D-MuPPET( pigeons in wild) and 3D-SOCS (great tits in semi-wild). Both contain 3D, keypoint based pose for birds (pigeons, tits) with multi-view cameras. The sequence also shows one or more birds, these two datasets seem to be ideas for such method since both have multiple cameras. The authors have not mentioned them, moreover, it would be interesting to know if the method can be applied to these datasets since all of these are specifically designed for behavior studies with 3D posture (i.e. gaze, etc.), also some are validated with motion capture with very high accuracy. Are these not suitable because of camera requirements of the proposed method?
The same with OpenMonkeyStudio (cited by authors), it is unclear why these datasets are not useful?
- The metrics used for evaluation of the results are not explained clearly. Only acronym is mentioned, this is assuming that all readers of the paper have in depth knowledge of the field. Keeping the broad scope of the paper explanation is necessary for a biologist in need of using the method in future. Also, some metrics e.g. R2, are not explained, unclear what the term.
- In addition, it would be helpful to specify how is the 3D position measured, a basic 3D prediction would mean XYZ based result, it is unclear to understand what is the final format of the result. This is crucial for users because in terms of behavioral experiments understanding pose sequence for behavior is one aspect but real world movement is also necessary in order to quantify movement of the animal in space (unless it is a head fixed animal), it is unclear if this can be obtained from the method?
- It seems the method, may not support multiple individuals at the same time. This is something crucial for multi-species experiments (indoor or outdoor), deserved to be mentioned in limitations (which are not explicitly detailed in the paper)

---

> ### Author Rebuttal · Authors · 2025-07-31
>
> Thank you for the detailed feedback and for highlighting the key strengths of our work. We appreciate your comment that we are tackling an important and challenging problem, and that our method offers an easier-to-adopt alternative to traditional keypoint or mesh approaches without requiring any annotations or species-specific information. Thank you also for noting our detailed explanation and the thorough validation across multiple datasets. We have addressed specific points in your review below.
>
> **Number of Cameras:** You are right that our use of the phrase “a small set of calibrated cameras” in the introduction is vague. While we provide the specifics in the experiments section, we will clarify in the introduction of our final version that our method was tested using 4 to 6 cameras and that we achieve best results with 5 or more cameras, as seen in Figure 5b.
>
> We chose a 6-camera configuration because it aligns with the standard established by Rat7M, which is one of the most widely-used benchmarks for this task. A 6-view arrangement is a common standard in modern neuroscience for achieving precise 3D kinematic tracking (e.g. Weinreb, C., Pearl, J. E., Lin, S., Osman, M. A. M., Zhang, L., Annapragada, S., ... & Datta, S. R. (2024). Keypoint-MoSeq: parsing behavior by linking point tracking to pose dynamics. _Nature Methods_, 21(7), 1329-1339.). We will add a sentence to the paper to provide this important context.
>
> Regarding two-camera stereo methods, our experiments with state-of-the-art models like PixelSplat and MVSplat highlight their core limitation for this task: they require substantial overlap between views to function effectively. This makes it difficult to reconstruct a complete 3D shape, as large portions of the animal's body are often occluded from any given pair of cameras. As our results show, their performance degrades significantly on our datasets where view overlap is sparse. Figure 12 further demonstrates that their performance does not improve even when given 3 or 4 views.
>
> Finally, regarding the practicality for large-scale experiments, a 4-6 camera system does not present a major barrier in a laboratory setting. Once the cameras are calibrated and fixed, they can collect longitudinal data for long durations and across different subjects with minimal effort. In outdoor or wild contexts, however, setting up this number of calibrated cameras may be prohibitive. We will add the following text to describe these limitations:
>
> > _Pose Splatter_ requires a minimum of roughly four calibrated cameras to operate, which may hinder its use in some outdoor or wild settings.
>
> **Multi-Animal Support:** You've correctly identified that our current model is designed and benchmarked for single-animal scenarios. As we note in the discussion, handling the "prolonged and severe inter-animal occlusions" in multi-animal scenes is a significant challenge that we have not yet benchmarked. While we mentioned this briefly as an area for future work, we will state it more explicitly in a limitations section in our revised manuscript:
>
> > Additionally, the model has no mechanism to handle occlusions of the animal, which can occur in complex environments and multi-animal scenarios.
>
> **Metrics (L1, IoU, PSNR, $R^2$, etc.):** We apologize for not defining the evaluation metrics more clearly. We omitted detailed explanations as they are standard in 3D reconstruction literature, but we agree that providing them would make the paper more accessible. We will add a new subsection to the revised manuscript defining each one.
>
> For your convenience, we provide a brief overview of the benchmark metrics used in our evaluation tables. To compare performance against other methods, we use four standard metrics. The **L1** metric measures the average absolute pixel-wise difference between our rendered image ($\hat{x}$) and the ground truth ($x$), evaluating color and texture accuracy, where a lower value is better. The **IoU (Intersection-over-Union)** score evaluates shape accuracy by calculating the overlap between the rendered silhouette ($\hat{m}$) and the ground-truth mask ($m$), where a higher score is better. We also report **PSNR** (peak signal-to-noise ratio) and **SSIM** (structural similarity index measure), which are standard image quality metrics where higher scores indicate a better, more perceptually similar reconstruction. It is worth noting that the L1 and IoU metrics also form the basis for our color loss ($L_{color}$) and silhouette loss ($L_{IoU}$), respectively, which are minimized during training.
>
> In our keypoint prediction experiments, we use the $R^2$ score, or the Coefficient of Determination. This metric measures the proportion of variance in the 3D keypoint coordinates that is predictable from our visual embedding. While the interpretation of what constitutes a ``good'' $R^2$ score is highly context-dependent, for complex and noisy biological data like animal behavior, a score above 0.5 generally indicates a strong predictive relationship. As shown in Figure 5 (left), our visual embedding proves to be a strong predictor for the animal's pose. For the finch, the $R^2$ values are consistently high (most > 0.75), indicating that our annotation-free embedding has captured the pose information almost as well as the manually derived keypoints themselves. For the mouse, we also see strong predictive power for most keypoints, demonstrating that our method successfully encodes quantifiable information about the animal's posture. This result is significant because it shows our embedding can serve as a rich, quantitative replacement for labor-intensive keypoint annotation in downstream analyses.
>
> **Other multi-view Datasets (3D-POP, 3D-MuPPET, 3D-SOCS):** Thank you for suggesting these valuable datasets. It seems like the single pigeon videos from 3D-POP could be a valuable addition to the paper. We also agree that adding experiments using the OpenMonkeyStudio data would be useful. We have begun processing this data and hope to add updates during the Author/Reviewer discussion period.
>
>
> **3D Position Output:** As stated in our methods section, the direct output of our network is a collection of 3D Gaussian particles representing the animal’s complete shape and appearance, which can be projected onto a 2D plane to create a rendered image from any given camera view. From a practical standpoint, the mean of each of these individual particles provides a real-world XYZ coordinate, and the full set of these points forms a representation of the animal's 3D shape that can be used to quantify movement.
>
> Our focus in the paper is to quantify the pose of the animal without reference to animal location or orientation (azimuthal angle), both of which are estimated in a pre-processing step described in Appendix A. _Pose Splatter_ uses these estimated values to produce predicted real-world 3D positions of the Gaussian particles that make up the animal.
>
> **Dataset Release and Documentation:** You are correct that the mouse and finch datasets are new contributions of this work. We are fully committed to open science, and as stated in the paper, the datasets will be released upon publication. We will include videos, mask videos, camera parameters, estimated 3D center and azimuthal angles, visual embedding time series, example code, and documentation. We are planning to release the data on the DANDI archive in Neurodata Without Borders (NWB) format. The DANDI archive is data repository geared toward enabling reproducible science within the neuroscience community, where each dataset is assigned a DOI.

---

> > ### Author Response · Authors · 2025-08-05
> >
> > We thank the reviewer again for pointing us to the 4-camera 3D-POP pigeon dataset and reminding us about the 64-camera OpenMonkeyStudio dataset. Unfortunately, we found that only labeled views of individual frames, and no videos from the OpenMonkeyStudio dataset were released, so we were unable to apply Pose Splatter. However, we are pleased to say that we have successfully applied Pose Splatter to the single pigeon 3D-POP data.
> >
> > **Adaptive Camera Parameters**
> > Compared to the three datasets tested previously (mouse, finch, and Rat7M), the cameras in the 3D-POP data are located further apart relative to the size of the animal. For this reason, we found that the shape carving procedure with the provided camera parameters sometimes failed to detect overlap among the back-projected masks. To correct for this, we applied an adaptive frame-by-frame adjustment to the intrinsic camera parameters.
> >
> > Briefly, we first find 2D mask centroids in each view and triangulate a rough 3D center of the animal using these centers and the provided camera parameters. We then re-project this 3D center point onto the 2D image planes and calculate a discrepancy between the reprojected point and the 2D mask centroids. Lastly, the camera center parameters ($c_x$ and $c_y$) are updated to remove the discrepancy.
> >
> > More specifically, we assume an intrinsic matrix of the form:
> > ```
> > [fx 0  cx]
> > [0  fy cy]
> > [0  0  1 ]
> > ```
> > We then project the 3D center point $x_\text{world}$ into camera coordinates: $x_\text{cam} = R x_\text{world} + t$, where $[R; t]$ are the camera's extrinsic parameters. Lastly, we update the intrinsic center parameters for each camera:
> >
> > $c_x \leftarrow u^* - f_x (x_\text{cam}^{(1)}/x_\text{cam}^{(3)})$
> >
> > $c_y \leftarrow v^* - f_y (x_\text{cam}^{(2)}/x_\text{cam}^{(3)})$
> >
> > where $u*$ and $v*$ are the image coordinates of the reprojected 3D center and $(x_\text{cam}^{(1)}, x_\text{cam}^{(2)}, x_\text{cam}^{(3)})$ are the three coordinates of the 3D center in camera coordinates.
> >
> > **Results**
> > Apart from this intrinsic parameter modification, which has been integrated into our codebase, all aspects of the pipeline remain unchanged for the pigeon data. The model achieves the following numbers on the test set (c.f. Table 2a, 4 cameras):
> >
> > | IoU$\uparrow$ | L1$\downarrow$ | PSNR$\uparrow$ | SSIM$\uparrow$ |
> > | ----- | ----- | ----- | ----- |
> > | 0.622 | 1.16 | 24.8 | 0.982 |
> >
> > Qualitatively, the renderings are of slightly lower quality than the mouse, finch, and rat 4-camera renderings, as expected from the lower PSNR, but still adhere to the overall shape of the pigeon. Unfortunately, we are unable to share renderings due to this year's NeurIPS discussion period guidelines.
> >
> > We plan to cite the 3D-POP dataset and add these results with representative renderings to the supplementary section of the paper. We are grateful for the opportunity to **demonstrate the effectiveness of Pose Splatter on a fourth species.**

---

> > > ### Comment · Reviewer_peFs · 2025-08-06
> > > **Comment to authors**
> > >
> > > Dear authors,
> > >
> > > Thanks for providing detailed points and also trying the experiment with existing datasets. The authors have addressed most of the queries. Although I disagree that 6+ camera for a single animals is a standard setup but this is not point of debate here. The method bring something very valuable to the table for biology community. The approach brings us step closer to full reconstruction in the wild for which datasets exist but methods do not.

---

### Note · Authors · 2025-08-11

Thank you to the reviewers for helpful comments and a productive discussion period. Below, we briefly summarize new results that will be included in the appendix of the paper based on reviewer feedback from the discussion period. These additional results reaffirm the strengths of Pose Splatter relative to alternative computer vision models.

* Quantitative and qualitative results on the **3D-POP single pigeon, 4-camera dataset**, demonstrating the effectiveness of Pose Splatter on a fourth species.
* Qualitative results applying the single-view method **BANMo**, showing inaccuracies in unobserved body parts.
* Qualitative results from the large pretrained 3D shape generation models **TRELLIS** and **HunYuan3D**, showing inaccurate shape generation for single-view conditioning.
* Qualitative results from multivew-conditioned **InstantMesh**, showing unrecognizable generated shapes.
* Qualitative results from **VGGT**, showing the "onion" layering effect of generated point clouds.
* Qualitative and quantitative results from **AnySplat**, both zero-shot and fine-tuned, showing the same onion-layer-like artifacts and inferior performance.

In addition, we thank the reviewers for their various suggestions to improve the clarity of the text and presentation, which we will incorporate into the paper.

---

### Decision · Program_Chairs · 2025-09-17

**Decision:**

Accept (poster)

**Comment:**

This paper introduces Pose Splatter, a feed-forward framework for reconstructing 3D animal pose and appearance from sparse multi-view recordings using Gaussian splatting. Unlike keypoint- or mesh-based pipelines, the method requires no annotations, templates, or per-video optimization, making it more scalable for large behavioral datasets. The authors also propose a rotation-invariant visual embedding derived from these reconstructions, which can replace 3D keypoints in downstream analyses. Results across mice, finches, rats, and pigeons show strong reconstruction quality and biologically meaningful embeddings, with clear efficiency gains over optimization-based methods.

Reviewers agreed that the core contribution lies less in novel network components and more in building a lightweight, practical system that works for challenging animal data where existing tools fail. The rebuttal was convincing in showing why large generalizable object or human-centric models (e.g., AnySplat, InstantMesh, TRELLIS) underperform in this setting, often hallucinating occluded geometry. Additional experiments on new datasets, including pigeons, strengthened the case for generalization, and clarifications around camera requirements, evaluation metrics, and limitations (single-animal support, multi-camera setups) improved accessibility. The release of code and datasets was also seen as a valuable community contribution.

While there was some debate about overstated claims and whether “pose estimation” is the right framing, the consensus is that this work addresses a timely and important challenge in neuroscience and animal behavior. It provides a practical tool that removes annotation bottlenecks and enables scalable longitudinal analyses. On balance, the reviewers converged toward acceptance: the paper is technically solid, well motivated, and makes a meaningful applied contribution that justifies inclusion at NeurIPS. I recommend acceptance.